# Synergy of ferroelectric polarization and oxygen vacancy to promote $CO_2$ photoreduction

Hongjian Yu[1], Fang Chen[1], Xiaowei Li[1], Hongwei Huang[1✉], Qiuyu Zhang[1], Shaoqiang Su[2], Keyang Wang[3], Enyang Mao[1], Bastian Mei [2], Guido Mul[2], Tianyi Ma [4✉] & Yihe Zhang[1✉]

Solar-light driven $CO_2$ reduction into value-added chemicals and fuels emerges as a significant approach for $CO_2$ conversion. However, inefficient electron-hole separation and the complex multi-electrons transfer processes hamper the efficiency of $CO_2$ photoreduction. Herein, we prepare ferroelectric $Bi_3TiNbO_9$ nanosheets and employ corona poling to strengthen their ferroelectric polarization to facilitate the bulk charge separation within $Bi_3TiNbO_9$ nanosheets. Furthermore, surface oxygen vacancies are introduced to extend the photo-absorption of the synthesized materials and also to promote the adsorption and activation of $CO_2$ molecules on the catalysts' surface. More importantly, the oxygen vacancies exert a pinning effect on ferroelectric domains that enables $Bi_3TiNbO_9$ nanosheets to maintain superb ferroelectric polarization, tackling above-mentioned key challenges in photocatalytic $CO_2$ reduction. This work highlights the importance of ferroelectric properties and controlled surface defect engineering, and emphasizes the key roles of tuning bulk and surface properties in enhancing the $CO_2$ photoreduction performance.

[1] Beijing Key Laboratory of Materials Utilization of Nonmetallic Minerals and Solid Wastes, National Laboratory of Mineral Materials, School of Materials Science and Technology, China University of Geosciences, Beijing, China. [2] Photocatalytic Synthesis Group, MESA+Institute for Nanotechnology, University of Twente, Enschede, The Netherlands. [3] The department of mechanics and engineering science, college of civil engineering and mechanics, Lanzhou University, Lanzhou, Gansu, P.R. China. [4] Centre for Translational Atomaterials, Swinburne University of Technology, Hawthorn, Victoria, Australia. ✉email: hhw@cugb.edu.cn; tianyima@swin.edu.au; zyh@cugb.edu.cn

CO$_2$ reduction into chemical fuels via a solar-energy conversion is an appealing and sustainable strategy towards utilization of renewable energy sources[1–5]. Photocatalytic processes for CO$_2$ conversion suffer from high activation energy barriers, rapid electron-hole recombination, and insufficient light utilization in most of the established photocatalysts[6–8]. In particular, the photocatalytic performance depends strongly on the photogenerated charge separation and transfer kinetics in the bulk and on the surface of photocatalysts[9]. As such, efforts have been made to improve the photocatalytic performance by using cocatalyst to increase the charge separation[10], surface modification strategies to enrich the reactive sites[11] and formation of heterojunction structures or facet junctions to enhance anisotropic photogenerated charge migration[12–14]. However, CO$_2$ conversion efficiencies are still low and not yet sufficient for potential industrial applications[15].

In ferroelectric semiconductors, the displacement of positive and negative charges allows for spontaneous polarization within the crystal; a strong driving force for bulk charge separation and among others, ferroelectric SrTiO$_3$[16] and BiFeO$_3$[17] have been reported to enhance the oxygen production performance of photoanodes. Ferroelectric materials are widely used as capacitors, piezoelectric sensors, and storage devices due to their unique spontaneous ferroelectric polarization. Although ferroelectrics have been proposed as potential photocatalytic materials starting from 2004[18], the research work up to now are generally focused on the influence of specific surface area or bandgap modification on photocatalysis[19,20].

Layered bismuth-based (LBB) semiconductor materials, as one of the most important classes of ferroelectrics, have lately gained considerable interest because of their unique structural properties that favor the formation of an internal static electric field for accelerating interlayer charge transfer. Our group has conducted pioneering work in this specific field. For instance, we demonstrated that ferroelectric polarization and in-turn charge separation was enhanced by post-annealing of ferroelectric SrBi$_4$Ti$_4$O$_{15}$ leading to increased CH$_4$ and CO production rates[21]. On the other hand, Li et al. revealed that the depolarization field a driving force to disperse photogenerated electrons and holes to the opposite polarization facets in single-domain ferroelectric PbTiO$_3$ nanoplates. Notably, the depolarization field can be directly increased with the increasing particle size along the polarization direction, further improving the photocatalytic hydrogen evolution reaction activity[22]. Despite the promising results obtained with polarized semiconductors, efforts in understanding and further development of this approach are mainly restricted in thin-film photoelectrodes. Employing effective polarizing routes like corona poling to strengthen the ferroelectric polarization of particulate photocatalysts is of great interest.

The interaction of photocatalysts and reactants is another important property of an efficient photocatalyst. For example, surface defects in oxygen-deficient BOCl have shown the ability to enable CO$_2$ adsorption and enhance CO$_2$ hydrogenation[23]. Our group prepared surface oxygen vacancies (OVs) modified Aurivillius phase Sr$_2$Bi$_2$Nb$_2$TiO$_{12}$ nanosheets leading to promoted CO$_2$ adsorption and conversion, proven by both computation simulation and experimental evidences[7]. Though the pioneering theoretical work using the two-dimensional Ising model further revealed that the existence of OVs in ferroelectrics might reduce ferroelectric properties (fatigue effect)[24], experimental proof and/or in-depth mechanistic understanding of the impact of OVs on ferroelectric domain and overall polarization in promoting photoinduced charge separation are still vacant.

In this study, the effect of ferroelectric polarization and surface OV concentration are explored in detail using Bi$_3$TiNbO$_9$ nanosheets as proof-of-concept materials, while the generated principle is expected to be extendable to another material system.

Surface OVs concentrations were adjusted and corona poling treatment was used to further enhance charge separation. The optimized treatment resulted in improved photocatalytic CO$_2$ reduction performance yielding CO production rates of 20.91 μmol g$^{-1}$ h$^{-1}$, outperforming pristine Bi$_3$TiNbO$_9$ nanosheets and those treated by single means. Experimental and theoretical results demonstrated that the photoresponse range, charge separation efficiency, and surface-active sites were improved by corona poling and OVs. Thus, our systematic analysis reveals that ferroelectric polarization and OVs synergistically promote photocatalytic CO$_2$ reduction. The presented strategy might trigger additional research about tackling pivotal steps in photo(electro) catalysis to further promote CO$_2$ conversion.

## Results

**Catalysts characterization.** Bi$_3$TiNbO$_9$ crystalizes in the non-centrosymmetric orthorhombic space group $A2_1am$ with unit-cell parameters of $a = 5.43$ Å, $b = 5.39$ Å, and $c = 25.05$ Å. It has a typical Aurivillius-type layered crystal structure formed with alternating [Bi$_2$O$_2$] and [TiNbO$_7$] perovskite slabs along the $c$ axis. Bi atoms are situated within the [TiNbO$_7$] slabs (Fig. 1a, b). Bi$_3$TiNbO$_9$ (BNT) nanosheets were synthesized by a hydrothermal route with NaOH as the mineralizer, followed by treatments with different amounts of glyoxal (BNT-OVX, X = 1, 2, 3) to introduce OVs (Fig. 1a). Then, Corona poling was used to modify BNT and BNT-OV2, denoted as BNT-P and BNT-OVP, respectively. As indicated by X-ray diffraction (Supplementary Fig. 1), the crystallographic structure is well, maintained; neither corona poling nor OV creation caused a phase change of Bi$_3$TiNbO$_9$ nanosheets. This is also consistent with the Raman results (Supplementary Fig. 2).

The morphology of as-prepared samples is analyzed by scanning electron microscopy (SEM) and atomic force microscope (AFM). The thickness of nanosheets is measured to be approximately 10–30 nm for all samples (Supplementary Fig. 3a–d and Supplementary Fig. 4a, b), and their specific surface area is about 19–21 m$^2$/g for these samples (Supplementary Fig. 5), further indicating that treatments are non-destructive to catalysts structure. The nanosheet structure of BNT and BNT-P is confirmed by transmission electron microscopy (TEM) (Supplementary Fig. 6a and Fig. 1c). It is noteworthy that both BNT and BNT-P show ferroelectric domains. These domains appear brighter for BNT-P indicating stronger ferroelectric polarization within the respective nanosheet. Diffraction patterns determined by selected area electron diffraction (SAED) of the [200] and [220] zone axis are highly visible and of regular order, revealing that BNT is well crystallized (Fig. 1c). Thus, the dominantly exposed facet of BNT is the {001} facet, according to the layered growth of BNT nanosheets along the $c$ axis. To understand the polarization at the atomic scale, spherical aberration-corrected scanning transmission electron microscopy (ACTEM) was employed to investigate the microstructured domains in BNT-OVP. Atomic-resolution annular bright-field scanning transmission electron microscopy (ABF-STEM) was conducted to survey the surface atomic structure of BNT-OVP (Fig. 1d), which shows a clear and uniform arrangement of Bi, Nb, and Ti atoms. Nb and Ti atoms occupy the central site of the octahedron in the perovskite [BiTiNbO$_7$] layer, and the direction of their displacement in the unit cell coincides with the direction of the spontaneous polarization. As Bi atoms are much heavier than Nb and Ti, the Bi atomic columns are darker than those of Nb and Ti. The relative displacements of the center Nb$^{5+}$ and Ti$^{4+}$ cation are presented by vectors pointing from the center of the octahedron to the corner of its four nearest neighboring Bi$^{3+}$ cations. The direction of Nb and Ti atoms displacement illustrates

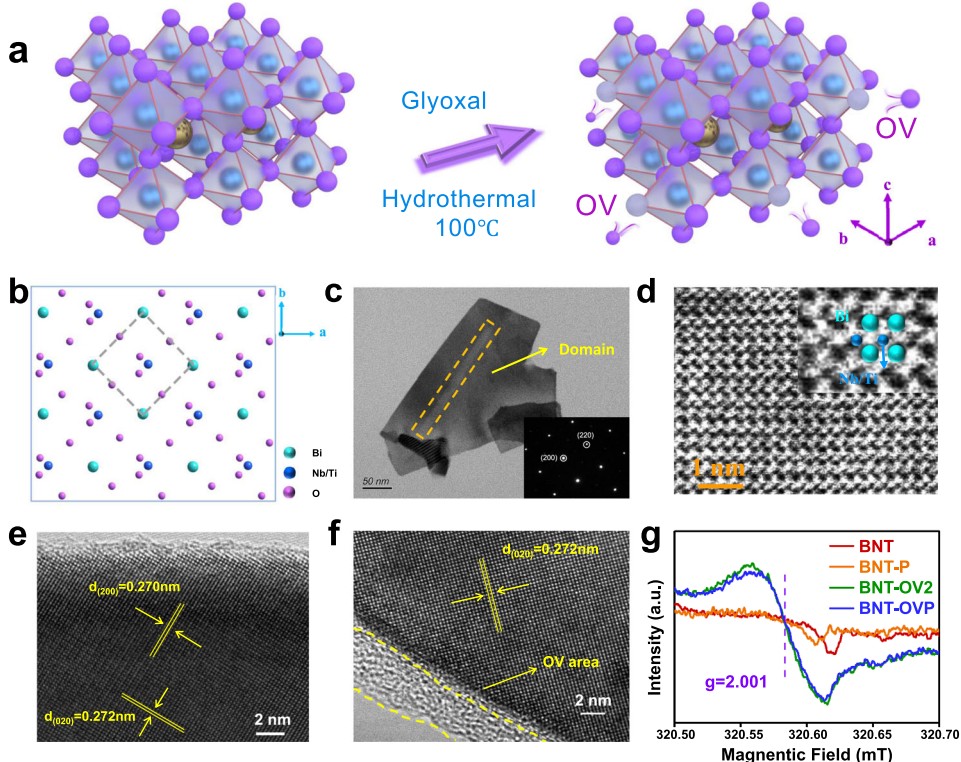

**Fig. 1 Structural and morphological information for BNT, BNT-P, BNT-OV2, and BNT-OVP. a** Schematic illustration for the formation of surface oxygen vacancies on $Bi_3TiNbO_9$. Blue, gold and purple spheres represent Ti/Nb, Bi and O atoms, respectively. **b** Crystal structure of $Bi_3TiNbO_9$. TEM image of (**c**) BNT-P, SAED pattern (inset) and (**d**) atomic-resolution ABF-STEM image of BNT-OVP. Inset: magnified view of (**d**) with corresponding Bi (cyan) and Ti/Nb (blue) columns. HRTEM images of (**e**) BNT and (**f**) BNT-OVP. **g** EPR spectra of BNT, BNT-P, BNT-OV2 and BNT-OVP.

a clear alignment of the Nb and Ti displacements for each unit cell along [110] direction, indicating a primary monodomain polarization along the [110] direction in BNT-OVP, further revealing the stronger polarization of BNT after corona poling. In sharp contrast to the neat crystal lattice of BNT (Fig. 1e), the thick amorphous edges and atomic defects in BNT-OVP suggest that the surface OVs mainly exist on the edges of the nanosheets[25,26] (Fig. 1f and Supplementary Fig. 6b). Still, energy dispersive X-ray (EDX) elemental mapping (Supplementary Fig. 7) confirms the homogeneous distribution of Bi, Nb, Ti, and O across the BNT-OVP nanosheets.

UV/vis diffuse reflectance spectra (DRS) demonstrate that BNT and BNT-P primarily absorb light below 400 nm, resulting in a bandgap of approx. 3.17 eV, consistent with the previous report (Supplementary Fig. 8)[27]. Furthermore, DRS measurements illustrate that light absorption remains unaffected by corona poling, whereas the absorption range was significantly extended after the introduction of OVs when comparing BNT-OV2 and BNT-OVP. Notably, there are two band gaps for BNT-OV2 and BNT-OVP, which are ~3.06 and 2.64 eV. This is consistent with the fact that OVs always cause tail absorption (Supplementary Fig. 9)[28,29]. Further, the presence of OVs was also confirmed by electron paramagnetic resonance (EPR) spectroscopy. In contrast to BNT and BNT-P, the typical signal at g = 2.001 was observed for BNT-OV2 and BNT-OVP, which is representative for electrons trapped in OVs (Fig. 1g)[23,28].

The surface composition and chemical states of related elements are first analyzed by X-ray photoelectron spectroscopy (XPS) (Supplementary Fig. 10a). For samples of BNT-OV2 and BNT-OVP subjected to reductive treatment, an evident shift (~0.15 eV) to smaller binding energies is observed in the high-resolution Nb 3*d* and Ti 2*p* core spectra compared with BNT and

BNT-P (Fig. 2a). This shift is indicative of a change in the chemical environment of Nb and Ti atoms present in sub-surface layers of the samples, i.e. the number of coordinating oxygen atoms is reduced by OV formation. Furthermore, the core spectra of bismuth (Bi 4*f*) reveal a less imperceptible shift (Supplementary Fig. 10b), suggesting that OV formation in $[Bi_2O_2]$ layers is less likely to occur during reductive treatment. The O 1*s* XPS spectra shown in Fig. 2b reveal the existence of lattice oxygen (529.9 eV) and surface hydroxyls (531.1 eV). In addition, a peak at 532.5 eV determined for BNT-OV2 and BNT-OVP can be assigned to adsorbed water, in agreement with a strong interaction between OVs sites with water vapor[7,30].

Extended X-ray absorption fine structure spectroscopy (EXAFS) of the Nb K-edge was performed to probe the local coordination environment. As shown in Fig. 2c, the Nb K-edge of BNT-OVP slightly shifts to lower energies, implying altered coordination of $Nb^{5+}$[31]. By further analyzing the Fourier transformed (FT) data of the extended fine structure region (Fig. 2d), an octahedral O environment (peak centered at R = 1–2 Å) and a reduction in Nb–O bond distance of Nb–O in BNT-OVP of 0.07 Å compared with BNT-P were revealed[32]. EXAFS wavelet transforms (WT) shown in Fig. 2e, f also highlight a shorter radial distance of Nb–O and a generally unsaturated coordination environment in BNT-OVP.

**Photocatalytic performance.** Photocatalytic $CO_2$ reduction performance is determined in a batch reactor using a gas-solid configuration. Simulated solar light was used for illumination; it was proven that only gaseous products were generated that can be detected by gas chromatography (Fig. 3a, b and Supplementary Fig. 11). Using pristine BNT, CO, and $H_2$ with production rates of

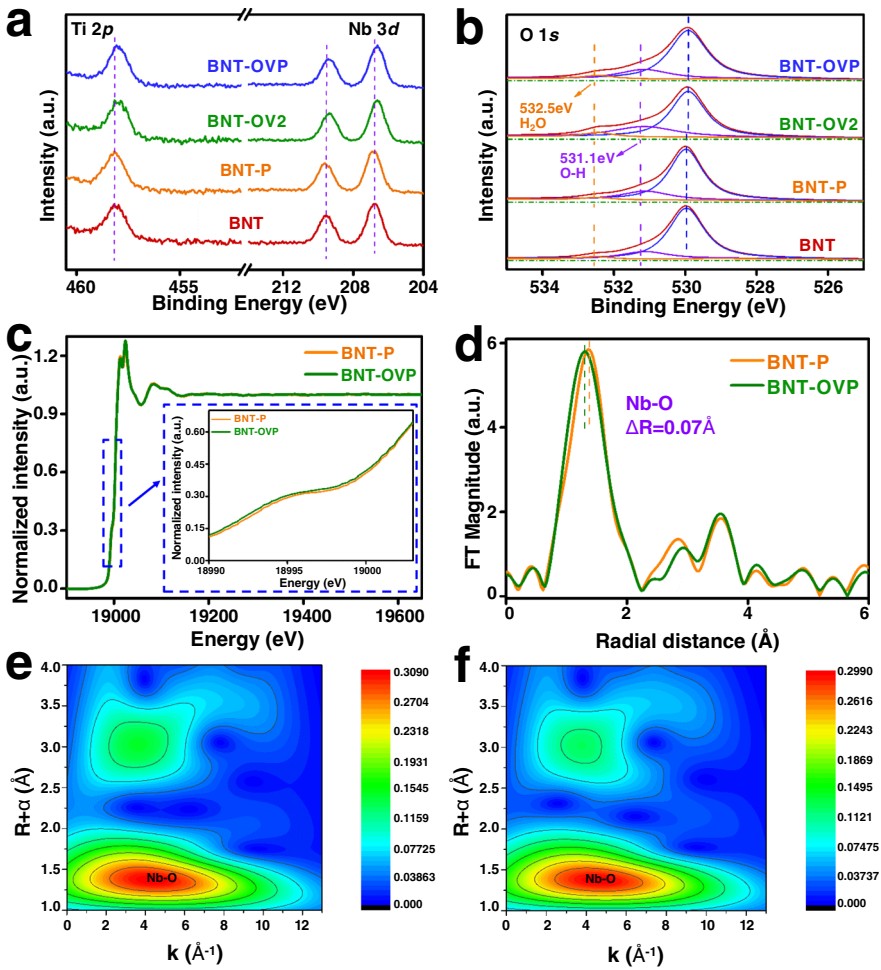

**Fig. 2 Characterization of the surface composition and chemical states of the samples.** XPS spectra of (**a**) Nd 3*d*, Ti 2*p*, and (**b**) O1s of BNT, BNT-P, BNT-OV2 and BNT-OVP. **c** Normalized Nb K-edge XAFS spectra and (**d**) Fourier transformed profiles for Nb coordination environments of BNT and BNT-OVP. ΔR represents the difference of Nb-O bond distance between BNT and BNT-OVP. **e**, **f** WT-EXAFS of BNT and BNT-OVP.

respectively 2.11 and 0.07 µmol g$^{-1}$ h$^{-1}$ were measured. BNT-P exhibits higher CO and H$_2$ yields with evolution rates of 10.25 and 0.15 µmol g$^{-1}$ h$^{-1}$, respectively. In addition, traces of CH$_4$ were detected (0.62 µmol g$^{-1}$ h$^{-1}$). Thus, corona poling increases the CO yield by ~5 times. Modification of the pristine BNT with OVs as in BNT-OVX (X = 1, 2, 3) also results in enhanced CO$_2$ photoreduction activity (Supplementary Fig. 12)[7]. The optimized CO and H$_2$ production rates of 5.29 and 0.11 µmol g$^{-1}$ h$^{-1}$ are determined for BNT-OV2. Corona poling of OV containing nanosheets (BNT-OVP) finally results in a CO evolution rate of 20.9 µmol g$^{-1}$ h$^{-1}$, exceeding the summed rates determined for BNT-P and BNT-OV2 (15.54 µmol g$^{-1}$ h$^{-1}$). This suggests that corona poling and OVs synergistically improve the CO$_2$ reduction activity of BTN-OVP, enabling a significant rate increase, even surpassing that of reported state-of-the-art photocatalysts under similar testing conditions (Supplementary Table 1), such as BiOCl (1.01 µmol g$^{-1}$ h$^{-1}$)[33], Sr$_2$Bi$_2$Nb$_2$TiO$_{12}$ with OVs (17.11 µmol g$^{-1}$ h$^{-1}$)[7], BiOIO$_3$ (17.33 µmol g$^{-1}$ h$^{-1}$)[34], and Br-Bi$_2$O$_2$(OH)(NO$_3$) (8.12 µmol g$^{-1}$ h$^{-1}$)[11]. In addition, the production rates of CH$_4$ and H$_2$ increase to 0.96 µmol g$^{-1}$ h$^{-1}$ and 0.19 µmol g$^{-1}$ h$^{-1}$, respectively. On the basis of the aforementioned results, the solar-to-CO/CH$_4$/H$_2$ conversion efficiency and the apparent quantum yield (AQY) with 365, 420, and 450 nm monochromatic irradiation for BNT-OVP are calculated to be about 0.021, 0.74, 0.46, and 0.35%, respectively. It is important to mention that in all cases the determined production rate of O$_2$

agrees well with the number of electrons required to enable the reductive processes to generate. CO, CH$_4$, and H$_2$ products (Fig. 3c).

In order to exclude any influence of organic impurities, blank experiments were conducted. Using Ar instead of CO$_2$ pre-purging, CO is detected only in traces (0.25 µmol g$^{-1}$ h$^{-1}$) (Supplementary Fig. 13); instead, a significant increase in H$_2$ production by water splitting is obtained. It is well known that effective suppression of the competing HER is essential to enable CO$_2$ photoreduction. As a matter of fact, CO$_2$ reduction is favorable over water splitting in CO$_2$ containing environment as proven in literatures[35]. Additionally, we confirm that in the absence of light and/or photocatalysts, no CO is formed (Supplementary Fig. 13). Using labeled carbon dioxide ($^{13}$CO$_2$) to trace the carbon sources, convincing proof is obtained that evolved CO and CH$_4$ indeed originate from the photoreduction of CO$_2$ (Fig. 3c). Notably, for BNT-P and BNT-OVP, the high durability of the polarization-induced electric field is revealed by the stable photocatalytic CO$_2$ reduction performance even after storage of one year (Supplementary Fig. 14). The durability of BNT-OVP is also confirmed for at least four consecutive reaction cycles (Supplementary Fig. 15–18). The favorable activity induced by both corona poling and OVs introduction is shown to be in good agreement with determined CO$_2$ adsorption capacities (Fig. 3d). In fact, CO$_2$ adsorption on BNT-OVP is well reflected by the sum CO$_2$ adsorption capacity of BNT-P and BNT-OV2.

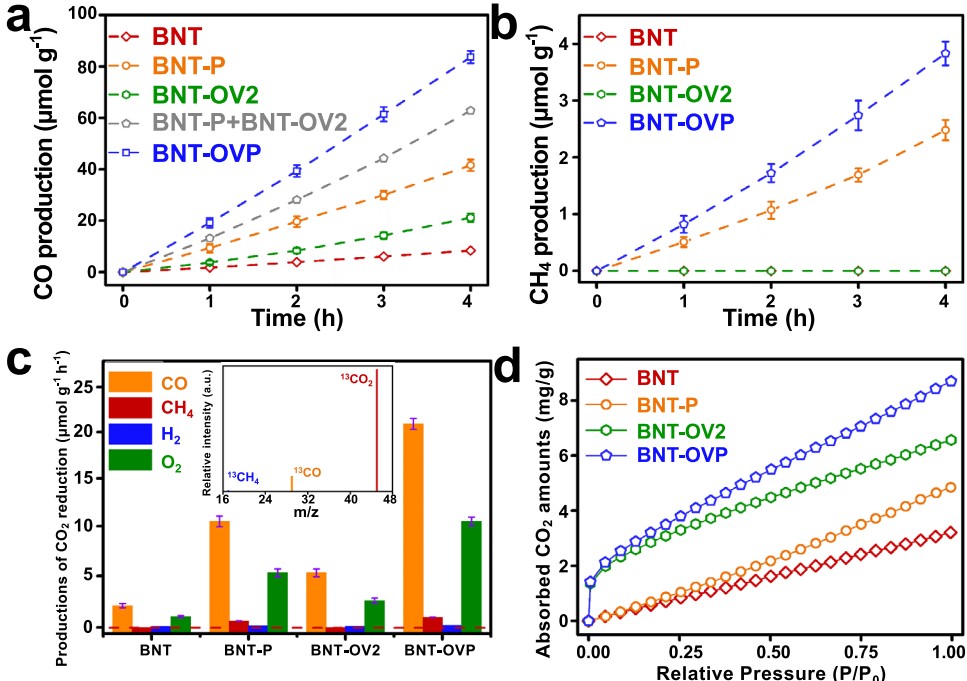

**Fig. 3 Photocatalytic performance. a, b** CO and CH$_4$ production curves and (**c**) corresponding evolution rates of CO, CH$_4$, and H$_2$ over BNT, BNT-P, BNT-OV2 and BNT-OVP under simulated solar light. Error bars represent the standard deviation. MS spectra for CO$_2$ reduction of BNT-OVP with using $^{13}$CO$_2$ as the reacting gas (inset). **d** CO$_2$ adsorption isotherms of BNT, BNT-P, BNT-OV2 and BNT-OVP.

Therefore, it is concluded that CO$_2$ adsorption on OVs is not affected by corona poling. Finally, *in-situ* FT-IR experiments indicate that CO$_2$ is adsorbed as carboxylate (CO$_2^-$, 1298 cm$^{-1}$), bidentate carbonate (b-CO$_3^{2-}$, 1381, and 1602 cm$^{-1}$), bicarbonate (HCO$_3^-$, 1205, and 1418 cm$^{-1}$), *HCOO (2883 cm$^{-1}$) and bidentate (2940 and 2981 cm$^{-1}$) that are eventually converted to CO and CH$_4$ upon illumination (Supplementary Fig. 19)[23,36]. The possible electron/proton transport processes and reaction pathways involved in the CO$_2$ reduction reaction over the as-prepared photocatalysts are proposed in Supplementary Fig. 20.

It has been reported that ferroelectric spontaneous polarization largely affects the charge separation of ferroelectric semiconductors[22,37]. The ferroelectric domains of Bi$_3$TiNbO$_9$ nanosheets are written and read out by Litho-PFM with a "CUGB" pattern. The writing voltage for the CUGB pattern is −10 V, and the rest is 10 V. From the hysteresis *P–E* loops (Fig. 4c and Supplementary Fig. 21a), BNT and BNT-OVP display a strong response of ferroelectric features and 180° phase switch. Besides, BNT-OVP presents remarkably higher remnant polarization and coercive field, which suggests the introduction of the corona polarization and OVs induces a stronger polarization electric field. The decreased saturation polarization causing fatigued ferroelectric characteristics reveals the influence of OVs on the dipole alignment[38]. The butterfly curves observed for BNT (Supplementary Fig. 21b) confirm its excellent piezoelectric properties. Atomic force microscopy (AFM) shows a clear CUGB pattern (Fig. 4a and Supplementary Fig. 22) attributed to the domain switching after applying switches of ±10 V in different zones of Bi$_3$TiNbO$_9$ nanosheets. Piezo-response force microscopy (PFM) illustrates the negatively and positively polarized domains at different voltages (Fig. 4b). Nevertheless, the unclear phase suggests unstable ferroelectric switching at low voltage (Supplementary Fig. 23). The obtained surface charge images show a heterogeneous charge distribution on the surface of BNT (Supplementary Fig. 24), in line with a polarization-induced electric field formed between bright and dark regions. These

results strongly suggest the presence of ferroelectric spontaneous polarization in Bi$_3$TiNbO$_9$ nanosheets, and that corona poling enhances the ferroelectric polarization, which results in more efficient charge separation.

It is remarkable that BNT-OVP displays an apparently larger difference between dark and bright regions and a higher surface charge potential (~45 mV) than that of BNT-P (~14 mV). OVs prevent back switching of ferroelectric domains after corona poling, exhibiting larger remnant polarization, which considerably promotes the charge separation of Bi$_3$TiNbO$_9$ nanosheets (Fig. 4d–f and Supplementary Fig. 25). The surface charge potential of BNT-OVP can be ascribed to the positive polarization and negative polarization with dark and bright regions due to the polarization-induced electric field (Supplementary Fig. 26). It is remarkable that BNT-OVP shows a ~5 mV decrease at the region of positive and negative potentials under the condition of illumination, which is caused by the separation and transferring to two opposite directions of the photogenerated electrons and holes. (Supplementary Fig. 26).

The relationship between charge separation efficiency, ferroelectricity, and surface oxygen vacancies is further analyzed by using photocurrent, time-resolved photoluminescence, and Mott-Schottky measurements. As shown in Fig. 4g and Supplementary Fig. 27 and 28, the highest current density of 3 μA cm$^{-2}$ is observed for BNT-OVP. Moreover, the current response at a longer wavelength observed for BNT-OV2 and BNT-OVP demonstrate that the formation of OVs indeed results in photon absorption at a higher wavelength, likely contributing to the enhanced CO$_2$ photoreduction activity. The average lifetime increases from 1.51 ns for BNT, 2.22 ns for BNT-P, 4.14 ns for BNT-OV2, to 7.33 ns for BNT-OVP, which indicates that BNT-OVP has the highest charge separation efficiency (Supplementary Fig. 29 and Supplementary Table 2). Similarly, the carrier density of BNT, BNT-P, BNT-OV2, and BNT-OVP is calculated to be $3.46 \times 10^{21}$, $4.44 \times 10^{21}$, $3.89 \times 10^{21}$, and $5.63 \times 10^{21}$, respectively (Fig. 4h and Supplementary Fig. 30), and the flat band

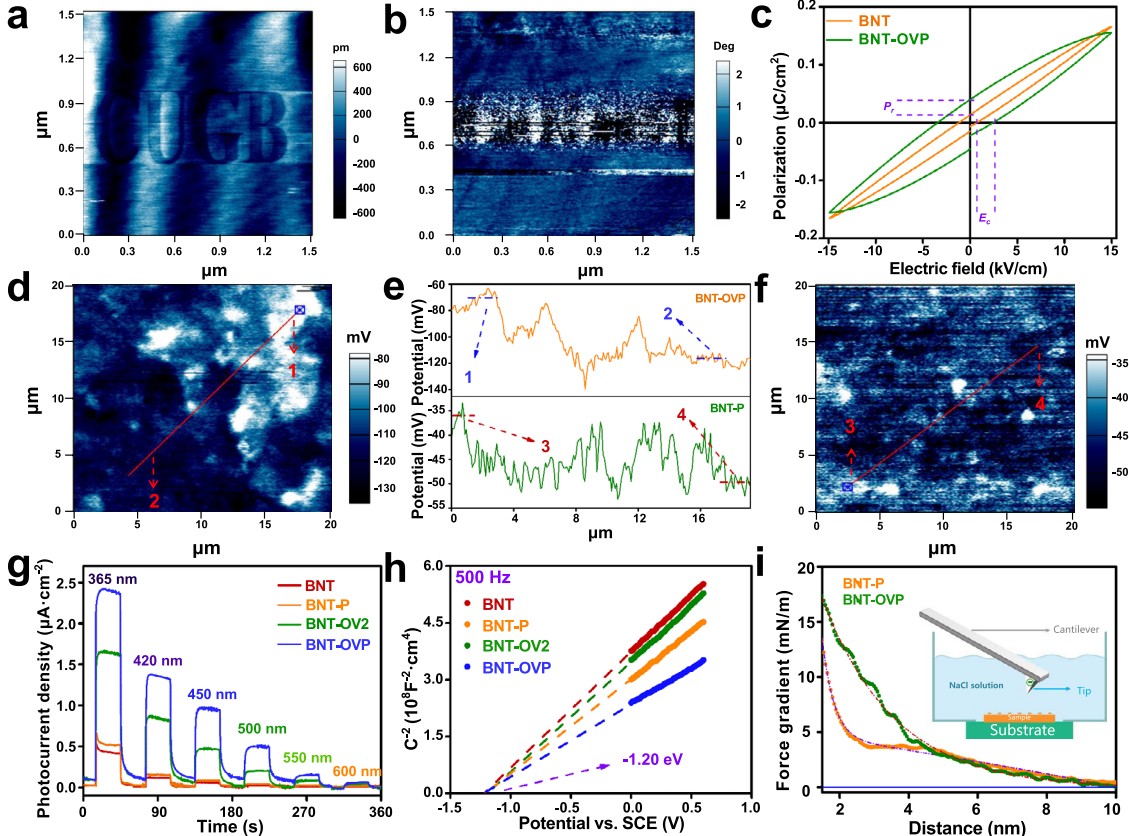

**Fig. 4 Effects of ferroelectric polarization and surface oxygen vacancies. a, b** AFM image and the standard ferroelectric phase image measured in air after applying +10 V and −10 V voltage of BNT. **c** polarization-electric field hysteresis loops of BNT and BNT-OVP. $P_r$ and $E_c$ represent remnant polarization and coercive field, respectively. **d, f** Surface charge and (**e**) corresponding charge difference profile of BNT-P and BNT-OVP. **g** Photocurrent response of BNT, BNT-P, BNT-OV2 and BNT-OVP under light with different wavelengths (**h**) Mott–Schottky plots of BNT, BNT-P, BNT-OV2 and BNT-OVP at a frequency of 500 Hz (0.1 M $Na_2SO_4$). **i** Combined 2D force field images, obtained by measuring 300 single force-versus-distance curves over BNT-P and BNT-OVP in 10 mM aqueous NaCl and averaged force curves from measurement in a plotted versus tip-sample distance.

potential was independent of corona poling and/or OV formation (Supplementary Fig. 31, 32). Meanwhile, AFM is employed to identify the charge density in the diffuse part of the electric double layer for BNT-P and BNT-OV2[39,40]. As shown in Fig. 4i, force-distance curves of BNT-P and BNT-OV2 both display the repulsive forces when the AFM tip made of oxidized Si with a negative charge is close to the surface. Compared with BNT-P, BNT-OVP shows an obvious larger repulsive force and a strongly negatively charged surface. These results further reveal the high charge separation efficiency of BNT-OVP caused by the synergistic effect of ferroelectric spontaneous polarization enhancement and the presence of OVs provides a stronger driving force (Supplementary Fig. 33).

To better understand the effect of OVs on the polarization properties, the system Hamiltonian and Poisson equations are employed based on a two-dimensional film of size $d_x \times d_z$ along the $x$ and $z$ directions. The change system Hamiltonian and spatial variation of the electric field caused by the central ion in the octahedral site displacing and space charge distribution along the $z$-direction is represented by Eqs. (1) and (2)[24,41].

$$H = -J \sum_{ij} \sum_{i'j'} S_{ij} S_{i'j'} - \boldsymbol{p_0} \sum_{ij} S_{ij} E^j + H_a + H_b \quad (1)$$

$$\frac{d^2\varphi}{dz^2} = -q_r f(z) \quad (2)$$

where $J$ is the coupling coefficient between neighboring dipoles, $S_{ij}$ is the two-dimensional array of *pseudospins* of the film, $S_{i'j'}$ is the

*pseudospin* state of one of the neighboring dipoles, $p_0$ is the dipole moment of the central ion, $E^j$ is the electric field along with the $z$ directions and can be obtained from Eq. (2), $H_a$ is the energy flipping between the upward dipole and downward one in the presence of an oxygen vacancy, $H_b$ is the energy term from the couplings of this distorted cage with the neighboring dipoles, $\varphi$ is the electrical potential, and $q_r$ is the charge concentration, $f(z)$ is the distribution function of oxygen vacancies along the polarization direction ($z$-axis).

The displacement of the central ion in the ($x, z$) cell is denoted as $u_{xz}$, the associated dipole moment is $p_{xz}$, where $p_{xz} = qu_{xz}$, and $q$ is the electron charge. The overall polarization can be estimated from the following expression:

$$\boldsymbol{P} = \frac{\sum_{xz} \boldsymbol{p_{xz}}}{d_x d_z} \quad (3)$$

It is clear that the introduction of OVs impedes the displacement of the center ion increasing the coercive field ($E_c$), and the domain switching needs extra energy to allow the deviated ions to return to near their original positions, which maintains a larger polarization intensity after poling. These results provide a theoretical perspective of the synergistic effect of OVs and poling field in promoting the separation of photo-generated charge carriers.

The domain switching in $Bi_3NbTiO_9$ nanosheets induced by corona poling was analyzed by COMSOL simulations. The simulations provide evidence that domains tend to gradually switch to be aligned when exerting a poling voltage, inducing a

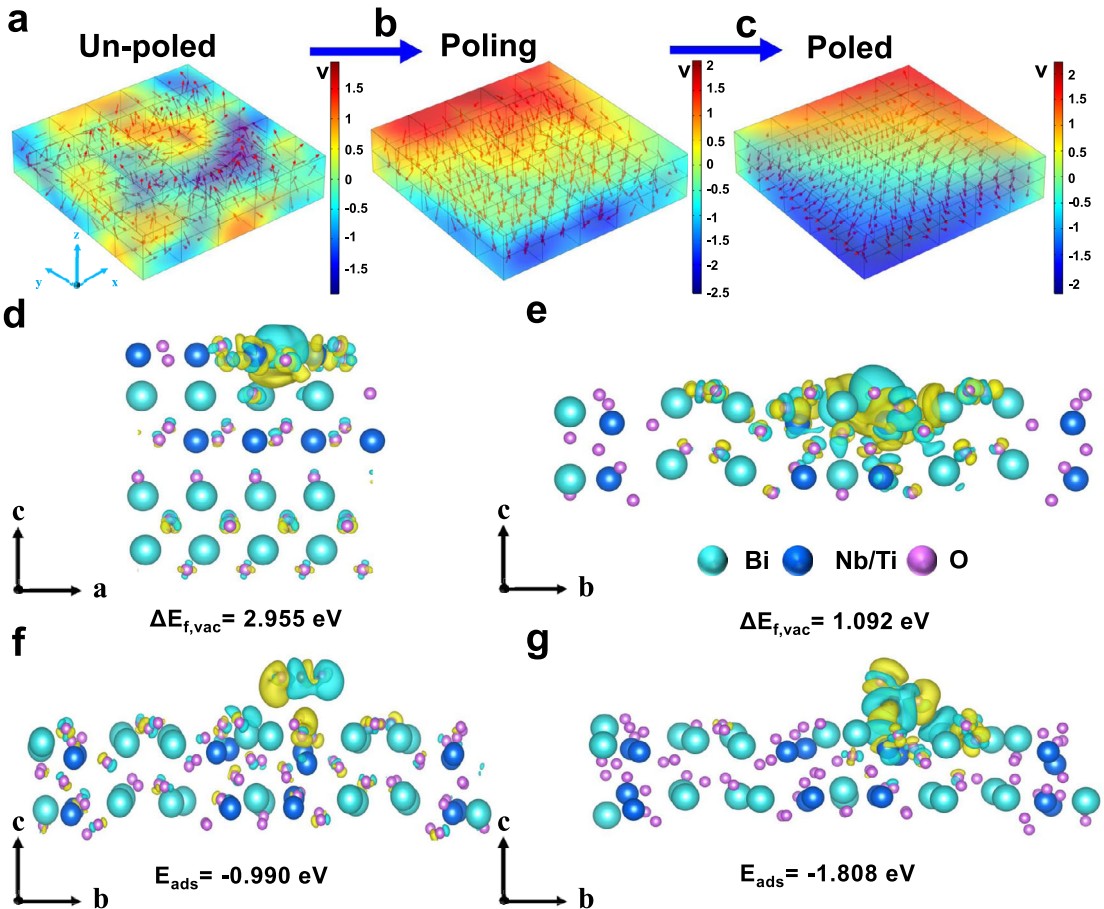

**Fig. 5 COMSOL simulations on Bi$_3$NbTiO$_9$ sheets under the process of corona poling and density functional theory (DFT) calculations.** COMSOL simulation of polarization-induced electric field on Bi$_3$NbTiO$_9$ sheets: (**a**) un-poled, (**b**) intermediate poled, (**c**) fully poled. The red arrow represents the polarization direction of a single domain. Formation energy ($\Delta E_{f,vac}$) of OV in the direction (**d**) perpendicular and (**e**) parallel to the ferroelectric polarization in the Nb/Ti-centered octahedral of Bi$_3$NbTiO$_9$with dipole moment correction. Charge difference of CO$_2$ adsorbed and adsorption energy ($\Delta E_{ads}$) on (**f**) Bi$_3$NbTiO$_9$ and (**g**) Bi$_3$NbTiO$_9$ with OVs (charge accumulation is in blue and depletion in yellow) with dipole moment correction.

strong electric field around the nanosheets (Fig. 5a–c). Thus the separation efficiency of photogenerated charge carriers can be improved with the increase of electric field intensity of spontaneous polarization under illumination. After the polarization voltage was removed, most domains trend to switch back, and a decrease of the polarization strength in the nanosheets can be observed. Introducing OVs are capable of retarding the back switching of domains, thus resulting in a higher persistent remnant polarization-induced electric field, as illustrated in Fig. 5c. More importantly, the positively polarized charge can bend down the energy band promoting the reduction reaction, while the negatively polarized charge can bend up the energy band enhancing the oxidation reaction[42], which can effectively improve the photocatalytic CO$_2$ reduction activity of Bi$_3$NbTiO$_9$ nanosheets. With the increase of poling voltage, the domains gradually switch to be aligned, resulting in a stronger potential difference caused by band bending (Supplementary Fig. 34a–c), which provides a stronger driving force for charge separation and higher photocatalytic activity.

It has been reported that ferroelectric polarization can lead to atom reconstruction on the surface of polar materials[43,44]. To get an in-depth understanding of the influence of ferroelectric spontaneous polarization on the position of formed OVs in Bi$_3$NbTiO$_9$, density functional theory (DFT) calculations were conducted towards OV formation energy ($\Delta E_{f,vac}$) of Bi$_3$NbTiO$_9$ at different positions. As shown in Supplementary Fig. 35, there

are dominantly two nonequivalent oxygen sites in the Nb/Ti-centered octahedra in Bi$_3$NbTiO$_9$, namely, one in the direction perpendicular and on in the direction parallel to the ferroelectric polarization, which are labeled as O$_I$ and O$_{II}$, respectively. The OV formation energy ($\Delta E_{f,vac}$) of O$_{II}$ (1.092 eV) is significantly lower than that of O$_I$ (2.955 eV), indicating that O$_{II}$ is easier to be absent to form OV (Fig. 5d, e), which is consistent with our HRTEM images. According to the Ising model and theoretical calculation, the OVs in the polarization direction will produce a pinning effect on the domain to hinder the switching of the domain, thus affecting the in-plane polarization and the separation of electrons and holes in this direction. Due to the existence of ferroelectric spontaneous polarization along [110], O$_{II}$ in BNT-OVP is more conducive to the formation of OVs to maintain stability when glyoxal is added. The oxygen vacancies defecting in the polarization direction eventually show a pinning effect on the domains, maintaining superb ferroelectric spontaneous polarization as a stronger driving force to greatly improve the separation efficiency of photogenerated carriers, which is the key to promote the catalytic performance of the photocatalyst.

To better understand the OVs' role on charge behavior and adsorption/activation processes of CO$_2$ molecules, the charge difference of Bi$_3$NbTiO$_9$ and Bi$_3$NbTiO$_9$ with OV for CO$_2$ adsorption is theoretically determined. In the presence of OVs, a much stronger charge interaction is observed between CO$_2$ and Bi$_3$NbTiO$_9$. The adsorption energy of CO$_2$ increases from 0.99 to

1.808 eV after the introduction of OV, further revealing a strong interaction between $CO_2$ and OVs (Fig. 5f, g), which is conducive to the following reduction processes.

## Discussion

In summary, $Bi_3TiNbO_9$ nanosheets were synthesized via a one-pot hydrothermal approach with the assistance of NaOH as a mineralizer. Subsequent modification by the creation of surface oxygen vacancies and treatment by corona poling resulted in a considerable improvement on the following crucial photocatalytic steps: (i) extending the photoresponse to the visible-light region; (ii) promoting the separation and migration of photoinduced electrons and holes to opposite directions; and (iii) providing a large number of reactive sites to effectively absorb and activate $CO_2$ molecules. Particularly, the OVs were demonstrated to show an imprint effect on the polarization-induced electric field to synergistically increase charge separation efficiency. Thus, the optimized $Bi_3TiNbO_9$ nanosheets with OVs treated by corona poling enabled $CO_2$ conversion with CO evolution rate of up to 20.91 µmol $g^{-1}$ $h^{-1}$, outperforming the majority of previously reported bismuth-based photocatalysts. This study suggests that photocatalytic properties can be favorably manipulated by ferroelectric polarization and surface defects. It is expected to inspire future research in the preparation of efficient photo-active materials.

## Methods

**Samples preparation**. All of the reagents used in this work were analytical grade and used as received without further purification.

The pristine $Bi_3TiNbO_9$ nanosheets (BNT) were synthesized by using $Bi_2(SO_4)_3$ as a bismuth source based on the process as follows: NaOH (Sinopharm) with a certain amount as mineralizer was dissolved in deionized water to achieve a NaOH concentration of 4 mol/L. Then, 1.5 mmol $Bi_2(SO_4)_3$ (Sinopharm) were added into the solution. After that, 1 mmol $Nb_2O_5$ (Sinopharm) and tetrabutyl titanate (Sinopharm) were added into the yellow turbid liquid. Then the mixture was transferred to a Teflon-lined autoclave (50 mL) and heated at 220 °C for 24 h. The as-prepared products (Supplementary Fig. 36) were washed with deionized water for several times and dried at 60 °C for 12 h in a vacuum drying oven.

Synthesis of oxygen-deficient $Bi_3TiNbO_9$ nanosheets: First, 200 mg $Bi_3TiNbO_9$ nanosheets are added into 30 mL glyoxal-water solution with ultrasonic agitation. And then the mixture was transferred to a Teflon-lined autoclave (50 mL) and heated at 100 °C for 6 h. The samples prepared with the amounts of glyoxal of 0.3, 0.5, and 1 mL are named as BNT-OV1, BNT-OV2, and BNT-OV3, respectively.

Corona poling process: 50 mg of BNT or BNT-OV2 powder was uniformly dispersed on a negative disk-like copper electrode (about 3 $cm^2$ in area). And the voltage of the steel-point electrode was 20 kV with a 1 cm distance between the two electrodes for 30 min to obtain polarized powder BNT-P or BNT-OVP, respectively (Supplementary Fig. 37).

**Samples characterizations**. The phase of as-prepared samples was recorded on an X-ray diffractometer (Bruker AXS, Germany) with Cu Kα radiation (λ = 1.5418 Å). Scanning electron microscopy (SEM, S-4800 Hitachi, Japan) was applied to identify the microstructure and morphology of obtained materials. Transmission electron microscopy (TEM) and high-resolution transmission electron microscopy (HRTEM) were conducted by FEI talos F200X and Tecnai F20 electron microscopy. UV-Vis spectrometer (Cary 5000 Varian, America) with $BaSO_4$ as the reference is used to obtain the optical absorbance spectra of the samples. Electron paramagnetic resonance (EPR) measurement was taken on an Endor spectrometer (ES-3X, JEOL, Japan). The surface composition and chemical states of the as-synthesized samples were analyzed based on X-ray photoelectron spectroscopy (XPS, ESCALAB 250 Xi ThermoFisher, UK). The X-ray absorption fine structure (XAFS) informations over the Nb K-edge of the photocatalyst were measured at the Advanced Phonon Source of Argonne National Laboratory at room temperature. $Bi_3TiNbO_9$ nanosheets by dry pressing were characterized with the Oxford MFP-3D AFM (Oxford MFP-3D, UK) under electric force microscope pattern (EFM) and piezoresponse force microscopy (PFM). Atomic Force Microscopy (AFM) (Bruker Icon) studies were performed to reveal the intrinsic surface charge of different samples. A negatively charged AFM-tip made from Si was used to scan over samples coated on a silica substrate in 10 mM aqueous NaCl. The ferroelectric property was probed through a ferroelectric tester (aixACCT Systems GmbH, Aachen, Germany). The Fourier transform infrared (FT-IR) spectra were analyzed by a Bruker spectrometer with the range from 400–4000 $cm^{-1}$ at room temperature. An automatic microporous physical and chemical gas adsorption analyzer (ASAP2020M PLUS) was used to measure the $CO_2$ adsorption isotherms of the samples.

**Photoelectrochemical tests**. Photocurrent and Mott-Schottky plots of samples were measured with the CHI-660E electrochemical system (Shanghai, China) using a three-electrode system. Detailedly, a saturated calomel electrode (SCE) was employed to be a reference electrode and a platinum wire act as counter electrode. The series of photocatalyst films were coated on indium-tin-oxide (ITO) glasses a working electrode. The electrolyte solution was 0.1 M $Na_2SO_4$ solution. The as-prepared samples were irradiated by a xenon lamp (300 W) with an AM1.5 G filter to achieve an intensity of 100 mW $cm^{-2}$. The measurements were conducted at room temperature at an open-circuit potential. The working electrodes were prepared using a dip-coating process. In short, 10 mg of photocatalysts were dispersed in1 mL of ethanol to be an even slurry. After that, the suspensions were dropped on 15 mm ×30 mm indium–tin oxide (ITO) glasses and dried at 373 K for 10 h to remove ethanol.

**Photocatalytic $CO_2$ reduction test**. The reduction products of $CO_2$ were detected using a closed circulation system (Labsolar-III AG, Beijing Perfect light Technology Co., Ltd., China) (Supplementary Fig. 38). 50 mg of obtained photocatalyst dispersed on a quartz dish and 1.3 g of $NaHCO_3$ were separately placed in the upper and lower in a reaction cell, which was then vacuum pumped. Subsequently, 5 mL of $H_2SO_4$ (4 M) was injected in the above reactor to react with $NaHCO_3$ for producing $CO_2$ gas (1 atm). Then, they were irradiated by a 300 W Xe lamp with an AM1.5 G filter (100 mW $cm^{-2}$ in intensity) with keeping the reactor temperature ∼ 20 °C. Then, 1 mL of the gaseous product was sampled and inspected using gas chromatography (GC9790II, Zhejiang Fuli Analytical Instrument Co.) with a thermal conductivity detector. Meanwhile, 1 mL of gas and 20 µL of liquid were detected by GC-7860Plus gas chromatography (Shanghai Yiyou Electronic Technology Co.) to analyze liquid products. 1 atm high purity Ar and 5 mL of water were injected into the reactor instead of 1.3 g of $NaHCO_3$ and 5 mL of $H_2SO_4$ to detect the possible influence of organic impurities on the surface of photocatalysts under the same experimental conditions. With the same experimental conditions, the light source was not turned on and photocatalyst was not added to the reactor in the blank experiment for dark and no photocatalyst, respectively.

**The calculation of solar-to-CO/$CH_4$/$H_2$ conversion efficiency (η) and apparent quantum yield (AQY)**. The solar-to-CO/$CH_4$/ $H_2$ conversion efficiency (η) was calculated as follows:

$$\eta(\%) = [R(CO) \times \Delta G°(CO) + R(CH_4) \times \Delta G°(CH_4) + R(H_2) \times \Delta G°(H_2)]/[P \times S] \times 100\%$$

where R(CO), R($CH_4$), R($H_2$), $\Delta G°$(CO), $\Delta G°$($CH_4$), $\Delta G°$($H_2$), P, and S denote the rate of CO, $CH_4$ and $H_2$ evolution (mol $s^{-1}$) in the $CO_2$ photoreduction system, the change in the Gibbs free energy that accompanies the reduction of $CO_2$ to CO (257 × 10³ J $mol^{-1}$) and $CH_4$ (818 × 10³ J $mol^{-1}$), the change of Gibbs free energy accompanying the water splitting (237 × 10³ J $mol^{-1}$), the solar light energy intensity (0.1 W $cm^{-2}$), and the illumination area (4 $cm^2$), respectively.

The solar-to-CO/$CH_4$/$H_2$ conversion efficiency (η) of BNT, BNT-P, BNT-OV2, and BNT-OVP were calculated to be about 0.0019%, 0.0111%, 0.0048%, and 0.0214%, respectively.

The monochromatic light of 365, 420, and 450 nm is employed to determine the apparent quantum yield (AQY) with the equations below:

AQY (%) = (number of reacted electrons/number of incident photons) × 100% = [(2 × number of evolved CO molecules + 8 × number of evolved $CH_4$ molecules)/number of incident photons] × 100%

The apparent quantum yield (AQY) of BNT-OVP were determined to be about 0.74%, 0.46%, and 0.35%, respectively.

**The calculated charge carrier densities**. The information of carrier density ($N_D$) in semiconductor photocatalysts can be reflected by Mott–Schottky plots based on the following equations:[45]

$$C^{-2} = \frac{2}{\varepsilon\varepsilon_0 q N_q}\left(E - E_{FB} - \frac{kT}{q}\right) \quad (4)$$

where C is the differential capacitance, ε represents the dielectric constant of semiconductor, $\varepsilon_0$ means the permittivity of vacuum, q is the elementary electron charge, E is the electrode potential, $E_{FB}$ is fiat band potential, k represents Boltzmann's constant, T is the absolute temperature, According to the linear relationship between C and E, the carrier concentration can be obtained from the slope of the straight line.

**COMSOL simulation of polarized electric field**. The geometrical characteristics were chosen as 100 nm × 100 nm × 20 nm, where the same domain is uniformly distributed. The space potential caused by domain interactions is approximately determined by the following equation:

$$\mathbf{D} = \varepsilon_0\mathbf{E} + \mathbf{P} \quad (5)$$

where D is electric displacement, $\varepsilon_0$ is the permittivity of vacuum, E is electric field strength, and P is polarization strength.

**Density functional theory (DFT) calculations**. To get in-depth cognition on the presence of oxygen vacancies in charge behavior enhanced adsorption and activation processes of $CO_2$ molecules, electronic band structure, structure relaxation, and single-point energies were investigated by the density functional theory (DFT) as implemented in the Vienna ab initio simulation package (VASP) with the generalized gradient approximation with the Perdew-Burke-Ernzerhof (PBE) exchange and correlation functional. Cut-off energy at 500 eV K-point sampling in the Brillouin area using the Monkhorst-Pack method was used to conduct the framework of the projector-augment wave. The convergences of energy and force were separately set as $1 \times 10^{-5}$ eV and 0.04 eV/Å, and the minimum density of the point is $2\pi \times 0.04$ Å$^{-1}$. The vacuum layer is set as 15 Å in the nonperiodic direction in the structural model to insure that the periodic mirrors have no interaction in this direction. The adsorption energy (Eads) for $CO_2$ molecules was determined by the following formula

$$E_{ads} = E_{tot} - (E_{mol} + E_{BNT}) \tag{6}$$

in which $E_{tot}$, $E_{mol}$, and $E_{BNT}$ represents the total energy of the adsorption structures, the energy of isolated molecules, and the energy of BNT structure, respectively.

## Data availability
The data that support the findings of this study are available from the corresponding author upon request.

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

## Acknowledgements

This work was jointly supported by the National Natural Science Foundations of China (No. 51972288, 51672258, 52071171), the Fundamental Research Funds for the Central Universities (292019145).

## Author contributions

H.J.Y. performed most of the experiments and wrote the first version of the paper; Y.H.Z. and H. W.H. co-supervised H.J.Y. regarding the work. Y.H.Z., H.W.H. and T.Y.M. co-initiated the research and co-revised the paper; F.C., X.W.L., and Q.Y.Z. carried out the DFT calculations; K.Y.W. help carry out the COMSOL simulation; S.Q.S. and E.Y.M. helped to analyze the results. B.M. and G.M. took part in the revision of the paper. All the authors discussed the results and commented on the paper.

## Competing interests

The authors declare no competing interests.
