## [Peer Review File · Nature Communications]

REVIEWER COMMENTS

Reviewer #1 (Remarks to the Author):

The article is written on an important topic, and I certainly think the authors have interesting experimental observations. Nevertheless, the theory section of the article is not thorough enough for publication at this stage. There are many caveats to the theory side of the article; below, I briefly mention a few of my most important concerns:

1) In Figure 5 it is shown that the ferroelectric polarization is in the in-plane direction. This is the direction in which the electron-hole separation is boosted by the ferroelectric polarization and electrons and holes get re-directed to different poles of the material. This being said the authors investigate the effect of oxygen vacancies on the surface of the material perpendicular to the z-direction (out-of-plane direction). Why should electron or hole transfer in this direction be influenced by the ferroelectric polarization?

2) The synergy between polarization direction and thermodynamic favorability of oxygen vacancies has not been discussed. There are several fundamental theory works on this subject in the literature. For example, see this:

<https://pubs.rsc.org/en/content/articlelanding/2016/ta/c6ta00513f#!divAbstract>

Oxygen vacancies or adatoms can appear in the direction parallel or anti-parallel to the ferroelectric polarization. This is a tunable quantity based on the magnitude of the polarization. But the ferroelectric polarization here is not parallel or antiparallel to the c-direction, so I don't know what makes the authors think that the oxygen vacancies are a controllable parameter here that has a synergy with ferroelectric polarization.

3) The surface reconstruction phenomena are not considered here. The surface of polar materials does not have a simple stoichiometry just like the bulk of the material. Rather the concentration of different kinds of vacancies or adatoms should be calculated by minimizing the surface Gibbs free energy. For example, see this:

<https://pubs.acs.org/doi/abs/10.1021/nl5035013>

Here the presence or absence of surface oxygens in the c-direction is not rationalized (based on thermodynamic calculations), neither the specific stoichiometry that the authors considered. When the authors create the oxygen vacancies, have they actually checked whether that surface corresponds to a stable phase in the surface phase diagram?

Reviewer #2 (Remarks to the Author):

Generally, this exploration article exhibited the synergetic impact between oxygen vacancies and ferroelectric properties. The author has been all around supported this idea; nonetheless, some idea content should be reexamined to look over it for detail clarification. The recently there are several numbers of the article have been published with a similar idea, so the author needs to consider in detail the clarification of peculiarity. The author also needs to underline the syndetic impact clarification, because when ferroelectric materials are annealed at vacuum condition, there may be conceivable to shape the oxygen vacancies. Hence, demonstrating this investigation with the synergetic impact, it may prompt another way. I recommend it as minor revision.

Comment1

The author mentioned that this material structure is nanosheets. So, is there any possibility that bandgap dependent on the number of layers? Besides, is nanosheets stacking effects on the activity and ferroelectric properties of $\text{Bi}_3\text{NbTiO}_9$ material? What does the author think about related CO_2 reduction activity? Besides, did you try to get the single layer of this nanosheet?

Comment2

Figure 1b shows that the perovskite structure sandwiched between two nanosheets. So, was there any issue of stability for the material? In the same figure 1g why is not much difference between BNT-OV2 and BNT-OVP magnetic field?

Comment3

In the case of photocatalytic activity batch reactor has used, and a source of CO_2 was $\text{NaHCO}_3/\text{H}_2\text{SO}_4$. So here, How has the author determined the specific concentration of CO_2 evolved from NaHCO_3 at a specific time?

The CO_2 reacted at 1atm on the surface of photocatalyst; however, at this pressure, the CO_2 adsorption it might not be good enough for photoreduction of CO_2 . So How authors deal with it?

During the photoreduction from Sodium bicarbonate, there might be the possibility of forming bi-dent or mon-dent carbonate interaction. Any similar author observed as if yes then add an explanation about it either not also.

SO_4^- ion it might find catalyst surface poisoning with Sulphur ion

The author has to show detailed optimization of a number of moles or concentration of CO_2 from a specific amount of NaHCO_3 and it.

In batch reactor at a specific amount of NaHCO_3 will produce a specific concentration of CO_2 and at a point after attaining the equilibrium, then there not be any more CO_2 . So How the author has shown that 4hrs reaction time. Because every hour's author has taken CO_2 gas from the batch reactor, at the time of initial reaction, the concentration of CO_2 will be different than during the 4hrs. As we can understand, after introducing the 1.3g NaHCO_3 and 5ml H_2SO_4 after 1hr irradiation time, a certain amount of CO_2 will be there but how we could say that the after 1hr also the same amount of CO_2 evolution is there.

Please do experiment with the 1.3g NaHCO_3 and 5ml H_2SO_4 fresh reaction mixture every hour and compared with current results.

Moreover, perform the TPD analysis for carbonate ion interaction with a photocatalyst, because here source of CO_2 is NaHCO_3

Comment4

The author has not explained about the selectivity of photoreduction $\text{Bi}_3\text{TiNbO}_9$. The after reaction between 1.3g NaHCO_3 and 5ml H_2SO_4 produces the H_2O in the reaction mixture and CO_2 to CO and low amount of CH_4 . Thus, it can conclude that there might be the possibility of high HER. Nonetheless, HER showed low, how it has been controlled reaction process and what is a specific reason, please added relevant explanation.

Moreover, in figure 3C in all samples, the H_2 evolution is the same if we compared the CO and CH_4 evolution, then corresponding why did not change in the HER activities? Please add a relevant explanation.

Comment5

In figure S3, there is not much difference in the specific area, and the case of BNT showed significant low activity concerning BNT-OV2 for CO evolution or even for CH₄. However, the specific area difference is not much. So, here the author focused surface engineering. So why is this contradiction? Please add a relevant brief description.

Comment6

As we mentioned earlier, in figure S15 cycling test, CO₂ concentration optimization, please add this part. Moreover, please include the information about the regeneration of catalyst after each cycle. Moreover, what happened after cycle 3, is that catalyst absolutely showed deactivation or was activated. If it showed the absolute deactivation, please add the relevant information and even included the SO₄⁻ poisoning because of Sulphur ion's consistent contact.

Comment7

In Mott-Schottky's analysis, figure 4h and S25, we are surprised that so all flat band potentials are similar to all samples? Please add an explanation. Furthermore, both Mott-Schottky analysis at different frequencies 500 and 800HZ for figure 4h and S25 respectively, but both analyses show the similar flat band potential. Please add specify the reason.

Comment8

Please check the possibility to analysis time-resolved photoluminescence spectroscopy to calculate the charge separation time for ferroelectric materials.

Comment9

In the figure, S26 add VB and CB specific values calculated from the Mott-Schottky analysis, which might explain the reaction mechanism. Moreover, write relatively reaction mechanism for CO, CH₄ and HER in brief.

Comment10

If possible, DFT calculation, then please perform DOS analysis for change in fermi level relatively changes in the oxygen vacancies. This will also help understand the reaction mechanism because as mentioned, selectivity for CO, CH₄ and HER can be proven.

Comment11

Please perform the electrochemical active surface area (ECSA) analysis and calculate the capacitance (C_s. μF/cm²). Moreover, add relevant information about the change in the electrochemical double-layer capacitance with relativity changes in the ferroelectric material's oxygen vacancies.

$$ECSA = C_{dl}/C_s$$

$$C_{dl} = (d(\Delta J))/(2dV_b)$$

Comment12

The Bi₃NbTiO₉ these are nanosheets so please perform the Raman spectroscopic analysis for charge density wave and magnetic transition in nanosheets.

Comment 13

Regarding OV, study the following related papers and cite them

- "Sustained, Photocatalytic CO₂ Reduction to CH₄ in a Continuous Flow Reactor by Earth-Abundant Materials: Reduced Titania-Cu₂O Z-Scheme Heterostructures", *Applied Catalysis B: Environmental*, 279 (2020) 119344.

- "A novel N-doped graphene oxide enfolded reduced titania for highly stable and selective gas-phase photocatalytic CO₂ reduction into CH₄: An in-depth study on the interfacial charge transfer mechanism", *Chemical Engineering Journal* (2020) <https://doi.org/10.1016/j.cej.2020.127978>

Reviewer #3 (Remarks to the Author):

In this manuscript, the authors prepared ferroelectric Bi₃TiNbO₉ (BNT) nanosheets via a mineralizer-assisted soft-chemical route. Surface oxygen vacancies were introduced to extend the photo-absorption and promote the adsorption and activation of CO₂ molecules on the catalysts' surface. Corona poling was further adopted to strengthen the ferroelectric internal polarization to facilitate bulk charge separation within the catalysts. Synergistic effects of surface oxygen vacancies and ferroelectric polarization were demonstrated in CO₂ photo-reduction. This manuscript is well organized with thorough characterization, therefore I recommend its publication in *Nature Communications* after some revisions as follows:

1. The authors stated that SEM images show the thickness of nanosheet to be 10-30 nm. Yet, it is difficult to confirm. Some additional evidence is suggested to be provided.
2. Will the corona poling process fill a part of the oxygen vacancies?
3. How does corona poling of BNT increase CO₂ absorption and thus lead to increased CO₂ reduction?
4. The authors stated "CO₂ is adsorbed as carboxylate (CO₂⁻ 221, 1298 cm⁻¹), bidentate carbonate (b-CO₃²⁻, 1381 and 1602 cm⁻¹), bicarbonate (HCO₃⁻ 222, 1205 and 1418 cm⁻¹), *HCOO (2883 cm⁻¹) and bidentate (2940 and 2981 cm⁻¹) that are eventually converted to CO and CH₄ upon illumination". How is CO₂ activated into these intermediates? What is the conversion route of these intermediates to CO or CH₄? Why CO is predominated in this study?
5. What is the isosurface level in Figure 3 e-g? It should be the same for comparison. The color of the atoms in these figures is suggested to be labeled. In addition, the authors stated that "In comparison with pristine BNT, the formation of OVs results in a higher charge accumulation in close proximity to the defect." However, obvious charge depletion was seen on the oxygen vacancy site in Figure 3e.
6. What is the mechanism on oxygen vacancy retarding the back switching of domains after poling? The authors stated that oxygen vacancy can retard the back switching of domains, thus leading to a higher remnant polarization. How does oxygen vacancy affect the origin ferroelectricity of BNT, as the literature has revealed that the existence of oxygen vacancies in ferroelectrics reduce the ferroelectric properties?
7. In Figure 5d, it is the un-poled sample, the "polarization" should be removed.
8. Much progress in CO₂ conversion has been made in recent years via adopting other methods on catalysts, reaction systems, etc. (e.g., *Adv. Funct. Mater.* 2020, 2005983; *Chemical Engineering Journal* 322, 22-32), which can be discussed.

06 June 2021

Dear Editor and Referees,

Thanks a lot for your comments and suggestions on our manuscript! Those suggestions are constructive and very helpful for revising and improving our paper. We have studied the comments carefully and made corrections accordingly. Please find below our point-by-point response.

Thanks for your kind consideration! I am looking forward to hearing from you.

With best regards,

Tianyi Ma | PhD | Professor | Fellow of RSC

Centre for Translational Atomaterials

Swinburne University of Technology

Mail No. H74 | PO Box 218, Hawthorn, VIC 3122, Australia

Room: AMDC 823 | Email: tianyima@swin.edu.au

Tel.: +61-3-92148625

Web Profile | Google Scholar | Publons

Reviewer #1

The article is written on an important topic, and I certainly think the authors have interesting experimental observations. Nevertheless, the theory section of the article is not thorough enough for publication at this stage. There are many caveats to the theory side of the article; below, I briefly mention a few of my most important concerns:

1. In Figure 5 it is shown that the ferroelectric polarization is in the in-plane direction. This is the direction in which the electron-hole separation is boosted by the ferroelectric polarization and electrons and holes get re-directed to different poles of the material. This being said the authors investigate the effect of oxygen vacancies on the surface of the material perpendicular to the z-direction (out-of-plane direction). Why should electron or hole transfer in this direction be influenced by the ferroelectric polarization?

Response: Thanks a lot for this comment! In Figure 5 of the original manuscript, we gave a schematic illustration of the domain switching in $\text{Bi}_3\text{TiNbO}_9$ in the simulated corona poling process. To obtain the polarization information in atomic level, in the revised submission we added spherical aberration corrected transmission electron microscopy (ACTEM) to investigate the microstructure of the domains in BNT-OVP. Atomic-resolution high angle annular bright-field scanning transmission electron microscopy was conducted to survey the surface atomic structure of BNT-OVP (Figure 1d), which shows clear and uniform arrangement of Bi, Nb and Ti atoms. Nb and Ti atoms occupy the central site of the octahedron in the perovskite $[\text{BiTiNbO}_7]$ layer, and the direction of their displacement in the unit cell coincides with the direction of the spontaneous polarization. As Bi atoms are much heavier than Nb and Ti, the Bi atomic columns are darker than those of Nb and Ti. The relative displacements of the center Nb^{5+} and Ti^{4+} cation are presented by vectors pointing from the center of the octahedron to the corner of its four nearest neighboring Bi^{3+} cations. The direction of Nb and Ti atoms displacement illustrates clear alignment of the Nb and Ti displacements for each unit cell along $[110]$ direction, indicating a primarily monodomain polarization state in BNT-OVP, further revealing the stronger polarization of $\text{Bi}_3\text{NbTiO}_9$ after corona poling. So the separation of photogenerated electron-hole pairs is boosted by the ferroelectric polarization along this direction. As such, Figure 5 has been updated in the revised manuscript.

With regard to the oxygen vacancies, it is found from the TEM images that they mainly appear at the edges of BNT-OVP nanosheets (the direction parallel or anti-parallel to the ferroelectric polarization), as shown in the Figure 1f. For further confirmation, we conduct the density functional

theory (DFT) calculations on the formation energy ($\Delta E_{f,vac}$) of oxygen vacancies at different sites in $\text{Bi}_3\text{NbTiO}_9$, which reveals that the oxygen vacancies are easier to be formed in the direction parallel to the ferroelectric polarization than that in the direction perpendicular to the ferroelectric polarization. The theoretical results will be detailedly discussed in our answers to the following two questions. Oxygen vacancies can exert a pinning effect on the domain and hinder the switching back of the domain after corona poling, which allow BNT-OVP to remain a larger polarization intensity for promoting the separation of electron-hole pairs in this direction. Please find below updated Figure 1 and corresponding discussions were added in the revised main manuscript.

Figure 1. Structural and morphological information for BNT, BNT-P, BNT-OV2 and BNT-OVP. (a) Schematic illustration for the formation of surface oxygen vacancies on $\text{Bi}_3\text{TiNbO}_9$. (b) Crystal structure of $\text{Bi}_3\text{TiNbO}_9$. TEM image of (c) BNT-P, SAED pattern (inset) and (d) Atomic-resolution ABF-STEM image of BNT-OVP. HRTEM images of (e) BNT and (f) BNT-OVP. (g) EPR spectra of BNT, BNT-P, BNT-OV2 and BNT-OVP.

2. The synergy between polarization direction and thermodynamic favorability of oxygen vacancies has not been discussed. There are several fundamental theory works on this subject in the literature. For example, see this: <https://pubs.rsc.org/en/content/articlelanding/2016/ta/c6ta00513f#!divAbstract>

Oxygen vacancies or adatoms can appear in the direction parallel or anti-parallel to the ferroelectric polarization. This is a tunable quantity based on the magnitude of the polarization. But the ferroelectric polarization here is not parallel or antiparallel to the c-direction, so I don't know what makes the authors think that the oxygen vacancies are a controllable parameter here that has a synergy with ferroelectric polarization.

Response: Thank you so much for this constructive comment and the suggested reference! We have carefully studied this reference, which inspires us to conduct the following investigations. The ferroelectric materials can attract electrically charged species from the ambient environment on their surfaces to screen the spontaneous electric field for the sake of charge neutrality. Different atomic vacancies are generated on both sides of the positive and negative polarization surface to keep the material stable, and the negative polarization surface tends to form oxygen vacancies. As described in our answer to your Question 1 above, the ACTEM image demonstrates that the ferroelectric polarization is along [110] direction (Figure 1d). The HRTEM images of BNT-OVP (Figure 1f) show that the surface oxygen vacancies mainly exist on the edges of the nanosheets, which demonstrate that the oxygen vacancies emerge in the direction parallel to the ferroelectric polarization, in good agreement with conclusion from the above literature.

As indicated by the reviewer, we further get insight into the synergy between polarization direction and thermodynamic favorability of oxygen vacancies; the formation energy ($\Delta E_{f,vac}$) of oxygen vacancies at different sites in $\text{Bi}_3\text{NbTiO}_9$ is also theoretically determined and compared (Figure 5d and e). The $\Delta E_{f,vac}$ of oxygen vacancies in the direction parallel to the ferroelectric polarization is 1.092 eV, which is significantly smaller than that in the direction perpendicular to the ferroelectric polarization (2.955 eV), which corroborates that the above experimental results. The oxygen vacancies as a controllable parameter eventually show an imprint effect on the domains in the polarization direction, which is the key to improve the charge separation and photocatalytic performance of the materials. Now, we have added the above discussions and the suggested important references in our revised manuscript.

Figure 5. COMSOL simulations on $\text{Bi}_3\text{NbTiO}_9$ sheets under the process of corona poling. (a, b) COMSOL simulation of polarization-induced electric field on $\text{Bi}_3\text{NbTiO}_9$ sheets: (a) un-poled, (b) intermediate poled, (c) fully poled. (d, e) Oxygen vacancy formation energy ($\Delta E_{f,\text{vac}}$) on $\text{Bi}_3\text{NbTiO}_9$ of axial and equatorial positions in the Nb/Ti-centered octahedral with dipole moment correction. Charge difference of CO_2 adsorbed on (f) $\text{Bi}_3\text{NbTiO}_9$ and (g) $\text{Bi}_3\text{NbTiO}_9$ with OVs (charge accumulation is in blue and depletion in yellow) with dipole moment correction.

3. *The surface reconstruction phenomena are not considered here. The surface of polar materials does not have a simple stoichiometry just like the bulk of the material. Rather the concentration of different kinds of vacancies or adatoms should be calculated by minimizing the surface Gibbs free energy. For example, see this: <https://pubs.acs.org/doi/abs/10.1021/nl5035013>*

Here the presence or absence of surface oxygens in the c-direction is not rationalized (based on thermodynamic calculations), neither the specific stoichiometry that the authors considered. When the authors create the oxygen vacancies, have they actually checked whether that surface corresponds to a stable phase in the surface phase diagram?

Response: To get in-depth understanding of the influence of polarization direction on the formation sites of oxygen vacancies, adsorbed CO_2 and surface reconstruction phenomena in $\text{Bi}_3\text{NbTiO}_9$, we have calculated the formation energy ($\Delta E_{f,\text{vac}}$) of oxygen vacancies at different sites in $\text{Bi}_3\text{NbTiO}_9$

and the adsorption energy of CO₂ on the surface of Bi₃NbTiO₉ and Bi₃NbTiO₉ with oxygen vacancies; the simulation is based on their energy-converged structural model with dipole moment correction by minimizing the surface Gibbs free energy. The much smaller $\Delta E_{f,vac}$ of oxygen vacancies in the direction parallel to the polarization (1.092 eV) than that in the direction perpendicular to the polarization (2.955 eV) demonstrates that the formation of oxygen vacancies parallel to the polarization direction is more thermodynamically favorable (Figure 5d, e), reflecting the stable phase, which is consistent with our experimental results. After dipole moment correction, it is notable that the oxygen vacancies and constituent atoms in Bi₃NbTiO₉ with oxygen vacancies shift slightly, while their composition and arrangement remain unchanged. It indicates that slight surface reconstruction occurs in this polar material, which will not obviously affect the photocatalytic activity. Besides, the adsorption energy of CO₂ on the surface of Bi₃NbTiO₉ is calculated to be -0.990 eV, which is smaller than that on Bi₃NbTiO₉ with oxygen vacancies (-1.808 eV), revealing that oxygen vacancies largely promote the adsorption of CO₂ molecules on the surface of this polar material.

Reviewer #2

Generally, this exploration article exhibited the synergetic impact between oxygen vacancies and ferroelectric properties. The author has been all around supported this idea; nonetheless, some idea content should be reexamined to look over it for detail clarification. The recently there are several numbers of the article have been published with a similar idea, so the author needs to consider in detail the clarification of peculiarity. The author also needs to underline the syndetic impact clarification, because when ferroelectric materials are annealed at vacuum condition, there may be conceivable to shape the oxygen vacancies. Hence, demonstrating this investigation with the synergetic impact, it may prompt another way. I recommend it as minor revision.

1. The author mentioned that this material structure is nanosheets. So, is there any possibility that bandgap dependent on the number of layers? Besides, is nanosheets stacking effects on the activity and ferroelectric properties of $\text{Bi}_3\text{NbTiO}_9$ material? What does the author think about related CO_2 reduction activity? Besides, did you try to get the single layer of this nanosheet?

Response: Thanks a lot for these good comments! For the nanosheet photocatalysts, the number of layers has a great influence on the bandgap, ferroelectric properties, photocatalytic activity. Compared with the bulk materials, thin-layer nanosheets always show a blue-shifted absorption edge and a larger bandgap. In addition, the thin-layer structure can increase the ferroelectricity by enhancing the polyhedral distortion, which strengthens the polarization electric field to promote the charge separation for enhancing the photocatalytic activity for CO_2 reduction. Thin-layer structure with more exposed surfaces can also provide abundant active sites for further enhancing the photocatalytic activity.^{1,2} Now, the corresponding explanations of the relationship between layer number and the physiochemical property of the materials were added in the revised manuscript together with relevant literatures.

It is indeed a good suggestion to synthesize single-layer nanosheet. However, we tried but failed to achieve this. It is because the high reaction temperature restricts the use of surfactants in the hydrothermal process of $\text{Bi}_3\text{NbTiO}_9$, where surfactant-induced method is the most established strategy reported. We will keep trying to find a suitable synthetic route to obtain monolayer of $\text{Bi}_3\text{NbTiO}_9$ for achieving higher photocatalytic performance in the future.

2. Figure 1b shows that the perovskite structure sandwiched between two nanosheets. So, was there any issue of stability for the material? In the same figure 1g why is not much difference between BNT-OV2 and BNT-OVP magnetic field?

Response: Generally, halide perovskites are unstable, but perovskite oxide is a stable material according to the previous reports and our experience. To confirm this, we compared the XRD patterns, XPS, EPR and cycling tests of photocatalytic CO₂ reduction of the catalysts before and after photoreaction; the results indicate that there is no obvious change in the crystalline phase, surface state, oxygen vacancies concentration or photocatalytic performance, which is in good agreement with the stable nature of this type of materials as claimed in literatures. These data have been provided in the Supplementary Figure 15-18. The corona poling process did not change the oxygen vacancies concentration and magnetism of the material, so the EPR signal between BNT-OV2 and BNT-OVP shows no much difference.

Supplementary Figure 17. (a) Survey XPS spectra, (b) Nd 3d and Ti 2p, (c) Bi 4f, and (d) O1s XPS spectra of BNT-OVP and BNT-OVP after photocatalytic CO₂ reduction.

3. In the case of photocatalytic activity batch reactor has used, and a source of CO₂ was NaHCO₃/H₂SO₄. So here, how has the author determined the specific concentration of CO₂ evolved from NaHCO₃ at a specific time?

The CO₂ reacted at 1atm on the surface of photocatalyst; however, at this pressure, the CO₂ adsorption it might not be good enough for photoreduction of CO₂. So how authors deal with it?

During the photoreduction from Sodium bicarbonate, there might be the possibility of forming bi-dent or mon-dent carbonate interaction. Any similar author observed as if yes then add an explanation about it either not also.

SO₄²⁻ ion it might find catalyst surface poisoning with Sulphur ion.

The author has to show detailed optimization of a number of moles or concentration of CO₂ from a specific amount of NaHCO₃ and it.

In batch reactor at a specific amount of NaHCO₃ will produce a specific concentration of CO₂ and at a point after attaining the equilibrium, then there not be any more CO₂. So how the author has shown that 4hrs reaction time. Because every hour's author hastaken CO₂ gas from the batch reactor, at the time of initial reaction, the concentration of CO₂ will be different than during the 4hrs. As we can understand, after introducing the 1.3g NaHCO₃ and 5ml H₂SO₄ after 1hr irradiation time, a certain amount of CO₂ will be there but how we could say that the after 1hr also the same amount of CO₂ evolution is there.

Please do experiment with the 1.3g NaHCO₃ and 5ml H₂SO₄ fresh reaction mixture every hour and compared with current results.

Moreover, perform the TPD analysis for carbonate ion interaction with a photocatalyst, because here source of CO₂ is NaHCO₃.

Response: We are very thankful for these good comment and concerns! The photocatalytic CO₂ reaction is conducted in a custom-designed reactor cell. As shown in Supplementary Figure 38, the as-prepared photocatalyst is dispersed evenly on the glass pane and placed on a triangle glass support in the 400 mL reactor cell; 5ml H₂SO₄ reacts with 1.3g NaHCO₃ to generate 400 mL CO₂ as the reaction source at the bottom of reactor cell. Here, the H₂SO₄ and NaHCO₃ do not contact the photocatalyst dispersed on the glass pane. The XPS results in Figure R1 show that there is no S peak detected in BNT-OVP, which excludes the presence of SO₄²⁻ on the surface of the catalyst. In the process of photocatalytic CO₂ reduction, 1 mL gas was extracted from the reactor cell per hour (4 ml in total in 4 hours). The volume of reactor is 400 mL, and the pressure in the reactor is only reduced by 1% after 4 hours, which is negligible. We also conduct the experiment of photocatalytic CO₂ reduction with the 1.3g NaHCO₃ and 5ml H₂SO₄ fresh reaction mixture every hour. As shown in Figure R2, the amount of evolved CO and CH₄ from CO₂ (umol/h) shows no obvious difference with that with the 1.3g NaHCO₃ and 5ml H₂SO₄ every 4 hours. According to the previous references, the pressure of photocatalytic CO₂ reduction reactors in the gas-solid reaction system is almost kept at 1 atm.³⁻⁵ In addition, the photocatalytic CO₂ reduction experiment by using

high-purity CO₂ gas and 5 ml H₂O as reactants instead of 1.3g NaHCO₃ and 5ml H₂SO₄ is also performed. The results show that the amount of evolved CO and CH₄ from CO₂ shows no evident difference for the different reactants (Figure R3). Furthermore, the CO₂-TPD test is conducted to investigate the interaction of CO₂ on BNT, BNT-P and BNT-OV2, BNT-OVP samples. As shown in Figure. R4, there are three major peaks, which can be assigned to weak (<200□), moderate (200□-400□), and strong (>400□) basic sites of catalysts, respectively. In general, the moderate and strong adsorption sites are active for CO₂ conversion, while the weak adsorption sites are inactive for CO₂ conversion. In comparison with BNT, BNT-P and BNT-OV2, BNT-OVP show stronger chemisorption of CO₂ and more moderate and strong basic sites, indicating that corona poling process and creating oxygen vacancies allow the CO₂ molecules to strongly interact with the catalyst and to be easily activated by the photoexcited electrons.

Figure R1. S 2p XPS spectra of BNT-OVP after photocatalytic CO₂ reduction.

Figure R2. The photocatalytic CO₂ reduction into CO (a) and CH₄ (b) over BNT-OVP with the 1.3g NaHCO₃ and 5ml H₂SO₄ fresh reaction mixture every hour.

Figure R3. Photocatalytic CO₂ reduction into CO (a) and CH₄ (b) over BNT-OVP with high-purity CO₂ gas and 5 ml H₂O.

Figure R4. CO₂-TPD profiles of BNT, BNT-P, BNT-OV2 and BNT-OVP.

4. The author has not explained about the selectivity of photoreduction $\text{Bi}_3\text{TiNbO}_9$. The after reaction between 1.3g NaHCO_3 and 5ml H_2SO_4 produces the H_2O in the reaction mixture and CO_2 to CO and low amount of CH_4 . Thus, it can conclude that there might be the possibility of high HER. Nonetheless, HER showed low, how it has been controlled reaction process and what is a specific reason, please added relevant explanation.

Moreover, in figure 3C in all samples, the H_2 evolution is the same if we compared the CO and CH_4 evolution, then corresponding why did not change in the HER activities? Please add a relevant explanation.

Response: The schematic illustration for electron/proton transport process and the formation of product is proposed according to the results of *in-situ* FT-IR experiments (Supplementary Figure 19)

to discuss the reaction mechanism of CO₂ reduction to CO and CH₄,^{6,7} which is added in Supplementary Figure 20. CO₂ molecules are easily adsorbed on the surface of Bi₃TiNbO₉ nanosheets, and then accept electrons and H⁺ to be converted into CO. The corona poling process enhances the ferroelectric properties of the samples, and the band bending allows the photo-generated electrons a higher reducing ability. Some CO molecules can accept two additional electrons, leading to carbon residue on the surface. These radicals can subsequently combine with up to four electrons and H⁺ eventually forming CH₄.

It is a reaction occurring at the gas/solid interface, and water vapor is the reactant in the process of photocatalytic CO₂ reduction and the corresponding oxidation product is O₂. There is no substantial liquid-phase water in the reaction system (vapour instead), which is the reason for low HER.⁸ Due to the low performance of HER, the change is not very obvious in Figure 3c. But actually, the H₂ evolution amount has increased from 0.07 μmol g⁻¹ h⁻¹ for BNT to 0.19 μmol g⁻¹ h⁻¹ for BNT-OVP as shown in Supplementary Figure 11.

Supplementary Figure 19. *In situ* FTIR spectra of BNT-OVP during CO₂ adsorption (0–30 min) and photoreduction (30–120min) processes in the range of 1200-3000 cm⁻¹.

Supplementary Figure 20. Schematic illustration for electron/proton transport process and the formation of product.

5. In figure S3, there is not much difference in the specific area, and the case of BNT showed significant low activity concerning BNT-OV2 for CO evolution or even for CH₄. However, the specific area difference is not much. So, here the author focused surface engineering. So why is this contradiction? Please add a relevant brief description.

Response: We are sorry for the misleading, and we have clarified our finding in the revised manuscript. Specifically, the surface engineering means creating oxygen vacancies, but not to mainly alter the specific surface area. Introduction of oxygen vacancies on the surface of Bi₃TiNbO₉ not only extends its photo-responsive range and tremendously promotes separation of photoinduced charge carriers, but also produces unsaturated bonds and dangling bonds for promoting the adsorption and activation of CO₂ molecules on the surface of the catalyst, thus greatly enhancing the photocatalytic activity. It is actually good that these samples have similar specific surface area, which can exclude the influence of specific surface area on the photocatalytic activity, further confirming the advantageous role of oxygen vacancies in promoting the photocatalytic CO₂ reduction. Corresponding clarifications were added in the updated version.

6. As we mentioned earlier, in figure S15 cycling test, CO₂ concentration optimization, please add this part. Moreover, please include the information about the regeneration of catalyst after each cycle. Moreover, what happened after cycle 3, is that catalyst absolutely showed deactivation or

was activated. If it showed the absolute deactivation, please add the relevant information and even included the SO_4^- poisoning because of Sulphur ion's consistent contact.

Response: Thank you for these good suggestions! We have performed the photocatalytic CO_2 reduction tests after three cycles. As shown in Figure R5, the CO and CH_4 evolution amount shows no notable decrease, which indicates that the photocatalyst still maintains high catalytic activity. After each cycling test, the photocatalyst after reaction was re-dispersed evenly on the glass pane and dried at $60^\circ C$ for 6 h for measurement. There is no special photocatalyst regeneration process. The XPS results in Figure R1 show that there is no S peak detected over BNT-OVP, which excludes the presence of SO_4^{2-} on the surface of the catalyst.

Figure R5. The cycling tests of photocatalytic CO_2 reduction into CO (a) and CH_4 (b) over BNT-OVP.

7. In Mott-Schottky's analysis, figure 4h and S25, we are surprised that so all flat band potentials are similar to all samples? Please add an explanation.

Furthermore, both Mott-Schottky analysis at different frequencies 500 and 800HZ for figure 4h and S25 respectively, but both analyses show the similar flat band potential. Please add specify the reason.

Response: To explain the unchanged flat band potential, density of states (DOS) of Bi_3TiNbO_9 is calculated by density functional theory (DFT) method. As shown in Supplementary Figure 32, the conduction band minimum (CBM) is composed of Ti 3d and Nb 3d orbitals, while the valence band maximum of Bi_3TiNbO_9 mainly consists of O 2p orbital, namely the O atoms do not contribute to CBM. As BNT, BNT-P, BNT-OV2 and BNT-OVP are all n-type semiconductors with a positive slope in Mott-Schottky plots (Figure 4h and Supplementary Figure 30), their flat band potential is very close to CBM. This is reason why the oxygen vacancies have little effect on the flat band potentials. The measured flat band potentials at different frequencies are consistent in theory.^{9,10}

Supplementary Figure 32. (a, b) Densities of states (DOS) of $\text{Bi}_3\text{TiNbO}_9$ and $\text{Bi}_3\text{TiNbO}_9$ with OV

8. Please check the possibility to analysis time-resolved photoluminescence spectroscopy to calculate the charge separation time for ferroelectric materials.

Response: According to referee's suggestion, we have added the test of time-resolved photoluminescence spectroscopy and calculate the charge separation time for these ferroelectric materials. As shown in Supplementary Figure 29 and Supplementary Table 2, the average lifetime is 1.51 ns for BNT, 2.22 ns for BNT-P, 4.14 ns for BNT-OV2 and 7.33 ns for BNT-OVP, respectively, which indicates that BNT-OVP has highest charge separation efficiencies.

Supplementary Figure 29. Time resolved PL curves for BNT, BNT-P, BNT-OV2 and BNT-OVP.

Supplementary Table 2. Time resolved PL decay parameter for BNT, BNT-P, BNT-OV2 and

BNT-OVP.

	τ_1 (ns)	A ₁	τ_2 (ns)	A ₂	τ_{avg} (ns)
BNT	0.47	16934.6	11.05	78.1	1.51
BNT-P	0.52	10152	11.07	91.4	2.22
BNT-OV2	1.02	4261	10.50	203.5	4.14
BNT-OVP	1.04	2511.1	12.62	246.1	7.33

9. In the figure, S26 add VB and CB specific values calculated from the Mott-Schottky analysis, which might explain the reaction mechanism. Moreover, write relatively reaction mechanism for CO, CH₄ and HER in brief.

Response: Thank you for this advice. The schematic illustration for electron/proton transport process and the formation of product has been added in Supplementary Figure 20 to discuss the reaction mechanism of CO₂ reduction to CO and CH₄.

10. If possible, DFT calculation, then please perform DOS analysis for change in fermi level relatively changes in the oxygen vacancies. This will also help understand the reaction mechanism because as mentioned, selectivity for CO, CH₄ and HER can be proven.

Response: We have added DOS of Bi₃TiNbO₉ and Bi₃TiNbO₉ with oxygen vacancies calculated by DFT in Supplementary Figure 32. The introduced oxygen vacancies lead to a new defect level in the band gap, which is beneficial to photoexcitation and charge separation. In addition, the schematic illustration for electron/proton transport process and the formation of product are added in Supplementary Figure 20 to discuss the reaction mechanism of CO₂ reduction to CO and CH₄.

11. Please perform the electrochemical active surface area (ECSA) analysis and calculate the capacitance (C_s. μF/cm²). Moreover, add relevant information about the change in the electrochemical double-layer capacitance with relativity changes in the ferroelectric material's oxygen vacancies.

$$ECSA = C_{dl}/C_s$$

$$C_{dl} = (d(\Delta J))/(2dVb)$$

Response: According to your suggestion, we have conducted the related electrochemical tests to analyze the electrochemical active surface area (ECSA) and calculate the capacitance (C_s , $\mu\text{F}/\text{cm}^2$). As shown in Figure R6 and Figure R7, C_{dl} is calculated to be 5.31, 5.37, 5.78 and 5.93 μF for BNT, BNT-P, BNT-OV2 and BNT-OVP, respectively. The results indicate that the introduction of oxygen vacancies can effectively promote the higher exposure of effective active sites of BNT-OV2 and BNT-OVP. The specific capacitance (C_s) of ITO substrate is calculated to be 8.28 $\mu\text{F}/\text{cm}^2$ (Figure. R8).

Figure R6. CV curves measured in a non-Faradaic region of 0.4-0.5 V at various scan rates for (a) BNT, (b) BNT-P, (c) BNT-OV2, and (d) BNT-OVP with a geometric area of 5 cm^2 , respectively.

Figure R7. Charging current differences ($\Delta I = I_a - I_c$) measured at 0.45V plotted against scan rate for BNT, BNT-P, BNT-OV2 and BNT-OVP, respectively. I_a and I_c are the anodic and cathodic current, respectively, and the linear slope is twice of the double-layer capacitance (Cdl).

Figure R8. Double-layer capacitance (C_{dl}) measurements for determining the specific capacitance (C_s) of FTO substrate from cyclic voltammetry (CV) in 0.1 M Na_2SO_4 : (a) CV curves measured in a non-Faradaic region of 0.4–0.5 V at various scan rates, (b) Charging current density differences ($\Delta J = J_a - J_c$) measured at 0.45 V plotted against scan rate. J_a and J_c are the anodic and cathodic current density, respectively, and the linear slope is twice of the C_s .

12. The $\text{Bi}_3\text{NbTiO}_9$ these are nanosheets so please perform the Raman spectroscopic analysis for charge density wave and magnetic transition in nanosheets.

Response: The Raman spectroscopic test at different temperatures has been conducted to distinguish the differences of metal-oxygen vibrations. In general, in layered Bi-based perovskite materials, the vibration modes below 200 cm^{-1} are assigned to the vibrations of the Bi^{3+} ions at

Bi₂O₂ layers or as A-site in perovskite slabs, and the modes above 200 cm⁻¹ are ascribed to the octahedral O-B-O and B-O vibrations (here A and B indicates the A and B sites in ABO₃ unit, respectively).^{11,12} As shown in Supplementary Figure 2, there is no obvious shift for the modes below 200 cm⁻¹, and the modes above 200 cm⁻¹ exhibit a slight shift to low-frequency position, indicating that oxygen vacancies occurred in the perovskite [TiNbO₇] slabs. As the temperature increases, some weak modes vanish especially in the wave number range of 200-900 cm⁻¹, which implies that its crystal structure and magnetic transition changes and the structural symmetry is improved. This phenomenon is attributed to the subtle change of crystalline phase from the ferroelectric A2₁am to an intermediate symmetry as the number of Raman modes decrease upon heating.¹¹

Supplementary Figure 2. Raman spectra of BNT and BNT-OVP at different temperatures.

13. Regarding OV, study the following related papers and cite them

- "Sustained, Photocatalytic CO₂ Reduction to CH₄ in a Continuous Flow Reactor by Earth-Abundant Materials: Reduced Titania-Cu₂O Z-Scheme Heterostructures", *Applied Catalysis B: Environmental*, 279 (2020) 119344.

- "A novel N-doped graphene oxide enfolded reduced titania for highly stable and selective gas-phase photocatalytic CO₂ reduction into CH₄: An in-depth study on the interfacial charge transfer mechanism", *Chemical Engineering Journal* (2020) <https://doi.org/10.1016/j.cej.2020.127978>

Response: Great thanks for your suggestion. We have carefully studied the above important references and cited them in our revised manuscript. Also, corresponding comparison between these literature and our current paper was added.

References:

- 1 Cheema, S. S. et al. Enhanced ferroelectricity in ultrathin films grown directly on silicon. *Nature*, **580**, 478–482(2020).
- 2 Xiong, J. et al. Ultrathin structured photocatalysts: A versatile platform for CO₂ reduction. *Appl. Catal. B-Environ*, **256**, 117788(2019).
- 3 Jiao, X. C., et al. Defect-mediated electron–hole separation in one-unit-cell ZnIn₂S₄ layers for boosted solar-driven CO₂ reduction. *J. Am. Chem. Soc.* **139**, 7586–7594 (2017).
- 4 Jiang, Z. F., et al. A hierarchical Z-scheme α -Fe₂O₃/g-C₃N₄ Hybrid for enhanced photocatalytic CO₂ reduction. *Adv. Mater.* **30**, 176108(2018).
- 5 Ye, L. Q., et al. Thickness-ultrathin and bismuth-rich strategies for BiOBr to enhance photoreduction of CO₂ into solar fuels. *Appl. Catal. B-Environ.* **187**, 281-290 (2016).
- 6 Habisreutinger, S. N. Schmidt-Mende, L. & Stolarczyk, J. K. Photocatalytic Reduction of CO₂ on TiO₂ and Other Semiconductors. *Angew. Chem. Int. Ed.* **52**, 7372-7408 (2013).
- 7 Kang, Q. et al. Photocatalytic Reduction of Carbon Dioxide by Hydrous Hydrazine over Au–Cu Alloy Nanoparticles Supported on SrTiO₃/TiO₂ Coaxial Nanotube Arrays. *Angew. Chem. Int. Ed.* **127**, 855-859 (2015).
- 8 Ang, L. et al. Three-Phase Photocatalysis for the Enhanced Selectivity and Activity of CO₂ Reduction on a Hydrophobic Surface. *Angew. Chem. Int. Ed.* **58**, 14549-14555 (2019).
- 9 Yin, X. F. et al. Realizing selective water splitting hydrogen/oxygen evolution on ferroelectric Bi₃TiNbO₉ nanosheets. *Nano Energy* **49**, 489-497 (2018).
- 10 Zhou, M. et al. Boron carbon nitride semiconductors decorated with CdS nanoparticles for photocatalytic reduction of CO₂. *ACS Catal.* **8**, 4928–4936 (2018).
- 11 Zhang, J. Z. et al. Lattice dynamics, dielectric constants, and phase diagram of bismuth layered ferroelectric Bi₃Ti_{1-x}W_xNbO_{9+ δ} ceramics. *J. Am. Ceram. Soc.* **99**, 3610–3615 (2016).
- 12 Sun, S. J. & Yin, X. F. Engineered layer-stacked interfaces inside Aurivillius-type layered oxides enables superior ferroelectric property. *Crystals* **10**, 710 (2020).

Reviewer #3

In this manuscript, the authors prepared ferroelectric $\text{Bi}_3\text{TiNbO}_9$ (BNT) nanosheets via a mineralizer-assisted soft-chemical route. Surface oxygen vacancies were introduced to extend the photo-absorption and promote the adsorption and activation of CO_2 molecules on the catalysts' surface. Corona poling was further adopted to strengthen the ferroelectric internal polarization to facilitate bulk charge separation within the catalysts. Synergistic effects of surface oxygen vacancies and ferroelectric polarization were demonstrated in CO_2 photo-reduction. This manuscript is well organized with thorough characterization, therefore I recommend its publication in Nature Communications after some revisions as follows:

1. The authors stated that SEM images show the thickness of nanosheet to be 10-30 nm. Yet, it is difficult to confirm. Some additional evidence is suggested to be provided.

Response: Thank you for this good suggestion! To further confirm the thickness of nanosheets, we have added the AFM images over BNT-OVP. As newly added in Supplementary Figure 4, the thickness of BNT-OVP is approx. 16 nm, which is consistent with the SEM results.

Supplementary Figure 4. AFM images and the corresponding height of BNT-OVP.

2. Will the corona poling process fill a part of the oxygen vacancies?

Response: Compared with BNT and BNT-OV2, the EPR signals of the poled BNT-P and BNT-OVP show no obvious change as shown in Figure 1g. Thus, the corona poling process will not fill the oxygen vacancy. In addition, there is no obvious change in surface state of elements (Figure 2a, b), further confirming that the corona poling process will not fill a part of the oxygen vacancies.

3. How does corona poling of BNT increase CO₂ absorption and thus lead to increased CO₂ reduction?

Response: The ferroelectric materials with polarized charges can attract electrically charged species from the ambient environment onto their surfaces to screen the spontaneous polarization electric field for the sake of charge neutrality. The corona poling endows BNT a stronger remnant polarization and more polarized charges, which gives rise to a better adsorption performance. As shown in Figure 3d, BNT-P and BNT-OVP show the stronger adsorption of CO₂. Furthermore, the CO₂-TPD test is conducted to investigate the interaction of CO₂ on BNT, BNT-P and BNT-OV2, BNT-OVP samples. As shown in Figure. R1, there are three major peaks, which can be assigned to weak (<200 °C), moderate (200 °C-400 °C), and strong (>400 °C) basic sites of catalysts, respectively. In general, the moderate and strong adsorption sites are active for CO₂ conversion, while the weak adsorption sites are inactive for CO₂ conversion. In comparison with BNT, BNT-P and BNT-OV2, BNT-OVP show stronger chemisorption of CO₂ and more moderate and strong basic sites, indicating that corona poling process and creating oxygen vacancies allow the CO₂ molecules to strongly interact with the catalyst and to be easily activated by the photoexcited electrons. It has been reported that the adsorption capacity of ferroelectric BaTiO₃ and Bi₄Ti₃O₁₂·(BiCoO₃)₂ for CO₂ and RhB increases with the increase of polarization intensity, which thus improves the catalytic performance of the materials.^{1,2}

Figure R1. CO₂-TPD profiles of BNT, BNT-P, BNT-OV2 and BNT-OVP.

4. The authors stated “CO₂ is adsorbed as carboxylate (CO₂⁻ 221, 1298 cm⁻¹), bidentate carbonate (b-CO₃²⁻, 1381 and 1602 cm⁻¹), bicarbonate (HCO₃⁻ 222, 1205 and 1418 cm⁻¹), *HCOO (2883 cm⁻¹) and bidentate (2940 and 2981 cm⁻¹) that are eventually converted to CO and CH₄ upon illumination”. How is CO₂ activated into these intermediates? What is the conversion route of these intermediates to CO or CH₄? Why CO is predominated in this study?

Response: Thank you for these good comments! The schematic illustration for electron/proton transport process and the formation of product is added in Supplementary Figure 20 to discuss the reaction mechanism of CO₂ reduction to CO and CH₄ according to the results of *in-situ* FT-IR experiments (Supplementary Figure 19).^{3,4} CO₂ molecules are easily adsorbed on the surface of Bi₃TiNbO₉ nanosheets with oxygen vacancies, and then further accept electrons and H⁺ to be converted into CO. The corona poling process enhances the ferroelectric properties of the samples, and the band bending makes the photo-generated electrons obtain higher reducing ability. Some CO molecules can accept two additional electrons, giving rise to carbon residue on the surface. These radicals can subsequently combine with up to four electrons and H⁺ eventually forming a small amount of CH₄.

Supplementary Figure 19. *In situ* FTIR spectra of BNT-OVP during CO₂ adsorption (0–30 min) and photoreduction (30–120 min) processes in the range of 1200–3000 cm⁻¹.

Supplementary Figure 20. Schematic illustration for electron/proton transport process and the formation of product.

5. What is the isosurface level in Figure 3 e-g? It should be the same for comparison. The color of the atoms in these figures is suggested to be labeled. In addition, the authors stated that “In comparison with pristine BNT, the formation of oxygen vacancies results in a higher charge accumulation in close proximity to the defect.” However, obvious charge depletion was seen on the oxygen vacancy site in Figure 3e.

Response: Thank you for this good comment. The isosurface level is 0.001. The color of the atoms in these figures has been labeled in our revised manuscript. The charge accumulation and depletion is in blue and in yellow, respectively. These corrections have made in the revised manuscript in Figure 5 d-g.

6. What is the mechanism on oxygen vacancy retarding the back switching of domains after poling? The authors stated that oxygen vacancy can retard the back switching of domains, thus leading to a higher remnant polarization. How does oxygen vacancy affect the origin ferroelectricity of BNT, as the literature has revealed that the existence of oxygen vacancies in ferroelectrics reduce the ferroelectric properties?

Response: Ferroelectric materials have attracted a great deal of attention for potential applications in ferroelectric random access memory, actuators and microwave electronic components. Fatigue

problems of ferroelectric materials, such as retention loss and imprint, refer to the reduction of switchable polarization after repetitive electrical cycling, which limit their commercial applications. Oxygen vacancies as typical defects in ferroelectric films can cause domain pinning, change domain structure and decrease reversible domains.^{5,6} Therefore, the presence of oxygen vacancies should be avoided in traditional ferroelectric thin films. In our work, we applied a high voltage on the samples by corona poling process to make the ferroelectric domains aligned. After removing the applied voltage, the domains of ferroelectric photocatalysts without oxygen vacancies are easier to deflect to near the initial position. Nevertheless, the introduction of oxygen vacancies will hinder this domains back switching process, which eventually retain greater remnant polarization.

7. In Figure 5d, it is the un-poled sample, the "polarization" should be removed.

Response: Thank you for this advice. Even if there is no corona poling process, the ferroelectric materials also have spontaneous polarization electric field, which will lead to band bending. So we think that it is necessary to keep the "polarization".⁷

8. Much progress in CO₂ conversion has been made in recent years via adopting other methods on catalysts, reaction systems, etc. (e.g., *Adv. Funct. Mater.* 2020, 2005983; *Chemical Engineering Journal* 322, 22-32), which can be discussed.

Response: Great thanks for your suggestion. The above important references have been cited in our revised manuscript. We have also add comparisons between our current work and these literatures in terms of performance and catalytic reaction mechanism.

References:

- 1 Li, D. B. , *et al.* Direct in situ determination of the polarization dependence of physisorption on ferroelectric surfaces, *Nat. Mater.* **7**, 473–477 (2008).
- 2 Li, X. N., *et al.* Enhancing oxygen evolution efficiency of multiferroic oxides by spintronic and ferroelectric polarization regulation. *Nat Commun* **10**, 1409 (2019).
- 3 Habisreutinger, S. N. Schmidt-Mende, L. & Stolarczyk, J. K. Photocatalytic Reduction of CO₂ on TiO₂ and Other Semiconductors. *Angew. Chem. Int. Ed.* **52**, 7372-7408 (2013).
- 4 Kang, Q. et al. Photocatalytic Reduction of Carbon Dioxide by Hydrous Hydrazine over Au–Cu Alloy Nanoparticles Supported on SrTiO₃/TiO₂ Coaxial Nanotube Arrays. *Angew. Chem. Int. Ed.* **127**, 855-859 (2015).

- 5 Pomyai, P. et al. Electrical fatigue behavior of $\text{Ba}_{0.85}\text{Ca}_{0.15}\text{Zr}_{0.1}\text{Ti}_{0.9}\text{O}_3$ ceramics under different oxygen concentrations. *J. Eur. Ceram. Soc.* **41**, 2497–2505 (2021).
- 6 Zou, X. et al. Mechanism of polarization fatigue in BiFeO_3 . *Acs Nano* **6**, 8997- 9004 (2012).
- 7 Liu, Y. et al. Internal-field-enhanced charge separation in a single domain ferroelectric PbTiO_3 photocatalyst. *Adv. Mater.* **32**, 1906513 (2020).

REVIEWERS' COMMENTS

Reviewer #1 (Remarks to the Author):

The authors have revised the manuscript according to my comments and those of other reviewers, and I think the manuscript is now ready for publication.

Reviewer #2 (Remarks to the Author):

The authors sincerely revised their work. I'm satisfied with the quality of the work. It can be published. Congrats.

Reviewer #3 (Remarks to the Author):

The comments raised are adequately addressed. The revised manuscript is recommended for publication in Nature Communications.